# Learning Audio-Visual Speech Representation by Masked Multimodal Cluster Prediction

**Bowen Shi**[1*]     **Wei-Ning Hsu**[2]     **Kushal Lakhotia**[2]     **Abdelrahman Mohamed**[2]

[1]Toyota Technological Institute at Chicago          [2]Meta AI

bshi@ttic.edu   {wnhsu,kushall,abdo}@fb.com

## Abstract

Video recordings of speech contain correlated audio and visual information, providing a strong signal for speech representation learning from the speaker's lip movements and the produced sound. We introduce Audio-Visual Hidden Unit BERT (AV-HuBERT), a self-supervised representation learning framework for audio-visual speech, which masks multi-stream video input and predicts automatically discovered and iteratively refined multimodal hidden units. AV-HuBERT learns powerful audio-visual speech representation benefiting both lip-reading and automatic speech recognition. On the largest public lip-reading benchmark LRS3 (433 hours), AV-HuBERT achieves 32.5% WER with only 30 hours of labeled data, outperforming the former state-of-the-art approach (33.6%) trained with a thousand times more transcribed video data (31K hours) (Makino et al., 2019). The lip-reading WER is further reduced to 26.9% when using all 433 hours of labeled data from LRS3 and combined with self-training. Using our audio-visual representation on the same benchmark for audio-only speech recognition leads to a 40% relative WER reduction over the state-of-the-art performance (1.3% vs 2.3%). Our code and models are available at https://github.com/facebookresearch/av_hubert

## 1 Introduction

Human perception of speech is intrinsically multimodal, involving audition and vision. The speech production is accompanied by the movement of lips and teeth, which can be visually interpreted to understand speech. Visual cues of speech not only play an essential role in language learning for pre-lingual children (Meltzoff & Moore, 1977; Davies et al., 2008), but also improve speech understanding in noisy environment (Sumby & Pollack, 1954) and provide patients of speech impairment with means of communication. Furthermore, perceptual studies (McGurk & MacDonald, 1976) have shown that such visual cues can alter the perceived sound.

For machine learning models, the tight coupling between audio and visual lip movement information emerges as a natural source for supervision to learn speech representations, which has not been extensively utilized yet in the self-supervised speech representation learning literature. Recent successful representation learning frameworks for speech (e.g., APC (Chung et al., 2019), CPC (Oord et al., 2018; Kharitonov et al., 2021), wav2vec 2.0 (Baevski et al., 2020; Hsu et al., 2021b), De-CoAR2.0 (Ling & Liu, 2020), HuBERT (Hsu et al., 2021c;a)) are mostly built entirely on audio. The fundamental research question addressed in this paper is whether a self-supervised audio-visual speech representation learned from the lip movement information, alongside the audio signal in video recordings, captures cross-modal correlations and improves downstream performance for visual speech recognition (i.e., lip reading) and automatic speech recognition (ASR) tasks. Existing ML models for lip-reading rely heavily on text transcriptions to achieve an acceptable level of accuracy. The state-of-the-art lip-reading model (Makino et al., 2019) requires 31K hours of transcribed video data for training. Such large amounts of labeled data are expensive and hard to obtain for most of the world's 7,000 languages. The benefit from a robust visual speech representation learning framework goes beyond lip-reading. Additionally, it can benefit a vast range of applications,

---
*Work done at Meta AI

including but not limited to keyword spotting in sign language (Albanie et al., 2020), speech enhancement (Xu et al., 2020) and talking face generation (Chen et al., 2018).

In this paper, we present Audio-Visual Hidden Unit BERT (AV-HuBERT), a multimodal self-supervised speech representation learning framework. It encodes masked audio and image sequences into audio-visual features via a hybrid ResNet-transformer architecture to predict the pre-determined sequence of discrete cluster assignments. The target cluster assignments are initially generated from signal processing-based acoustic features (e.g., MFCC) and iteratively refined using the features learned by the audio-visual encoder via k-means clustering. AV-HuBERT simultaneously captures linguistic and phonetic information for unmasked regions from both the lip-movement and audio streams into its latent representations, then encodes their long-range temporal relationships to solve the masked-prediction task.

The contextualized representations learned by AV-HuBERT show excellent transferability to the lip-reading task, where only the visual modality is available. Pre-training on audio and visual input streams led to substantially better results than only visual input. In the low-resource setup using only 30 hours of labeled data from LRS3 (Afouras et al., 2018b), our model achieves a lip-reading WER of 32.5%, outperforming the previous state-of-the-art model (33.6%) trained on 31,000 hours of transcribed videos (Makino et al., 2019). Using the complete 433 hours from LRS3 further reduces WER to 28.6%. We further show AV-HuBERT and self-training are complementary to each other: combining both sets a new lip-reading WER record of 26.9%. In addition, we show that the multimodal clusters derived from AV-HuBERT can be used to pre-train a HuBERT model for audio-based speech recognition, outperforming the previous state-of-the-art model (2.3%) and the unimodal HuBERT pre-trained on audio clusters (1.5%) by a large margin (1.3%).

## 2 RELATED WORK

The strong correlation between video modalities provides an effective means for self-supervised representation learning on videos, which has been explored in many prior works and is still an active research area. This work draws inspiration from two lines of previous research:

**Multimodal general video representation** focuses on learning self-supervised audio-visual representations of general videos to solve high-level semantic tasks, e.g., action recognition and audio event detection (Arandjelović & Zisserman, 2017; Bruno et al., 2018; Morgado et al., 2021; Chen et al., 2020; Lee et al., 2021). Owens & Efros (2018) learns a multimodal network to predict whether the audio and visual streams of a video are temporally synchronized while Pham et al. (2019) applies a cyclic translation between different modalities. Piergiovanni et al. (2020) learns the visual representation through a multi-tasking framework that incorporates a series of pretext tasks such as reconstruction and temporal ordering prediction. Sharing our work's inspiration of DeepClustering (Caron et al., 2018), XDC (Alwassel et al., 2020) and AV-BERT (Chan et al., 2021) learn cross-modal representations through predicting cross-modal or cluster assignments. In contrast to XDC, AV-HuBERT is trained with a BERT-like masked prediction loss, which forces the model to learn the structure within the multimodal input and was shown in Hsu et al. (2021c) to be more resilient to bad cluster assignments compared to unmasked cluster prediction. On the other hand, AV-BERT focuses on learning utterance-level multimodal environment embeddings that serves as the global context for ASR, while our objective is to learn frame-level audio-visual speech representations and pre-train a model that can be fine-tuned for downstream tasks with either modality.

**Semi- and self-supervised audio-visual speech representation learning** focuses on improving lip-reading with untranscribed audio-visual speech data. To solve isolated visual word recognition, Chung et al. (2020) learns visual embeddings using a contrastive loss based on audio-visual synchronization. Ma et al. (2021a) learns visual speech representations by minimizing the distance between its latent features and off-the-shelf audio embeddings. Using an external supervised ASR to transcribe unlabeled audio, Afouras et al. (2020) trains their lip-reading model on the augmented labeled and pseudo-labeled data. Unlike Ma et al. (2021a) and Afouras et al. (2020), our model is trained from scratch and encouraged to learn contextualized representations with a masked prediction task. Moreover, our method does not rely on any external pre-trained models.

## 3 METHOD

### 3.1 PRELIMINARY: AUDIO HUBERT

Our research builds on Audio HuBERT (Hsu et al., 2021a) which is a self-supervised learning framework for speech and audio. It alternates between two steps: feature clustering and masked prediction. In the first step, a discrete latent variable model (e.g., k-means) is applied to a sequence of acoustic frames $\mathbf{A}_{1:T}$ producing a sequence of frame-level assignments $\mathbf{z}^a_{1:T}$. Clusters of signal-processing-based acoustic features, e.g., Mel-frequency cepstral coefficients (MFCC), exhibit non-trivial correlations with the inherent acoustic units of speech inputs. Using $(\mathbf{A}_{1:T}, \mathbf{z}^a_{1:T})$ pairs, the second step learns new feature representations by minimizing a masked prediction loss, similar to masked language modeling in BERT (Devlin et al., 2019). The pressure to predict cluster assignments of masked audio regions forces the model to learn good local acoustic representations for unmasked regions and long-range temporal dependencies between latent features. Repeating these two steps improves the cluster quality and consequently the quality of the learned representations.

### 3.2 SINGLE-MODAL & CROSS-MODAL VISUAL HUBERT

**Single-modal Visual HuBERT:** The most naïve way to extend HuBERT to the visual domain is by generating targets using visual features. Formally, given an image sequence $\mathbf{I}_{1:T}$, we first cluster the image features into a sequence of discrete units $\mathbf{z}^i_{1:T}$ via k-means: $z^i_t = $ k-means$(G(\mathbf{I}_t)) \in \{1, 2, ..., V\}$, where $G$ is an visual feature extractor and $V$ is the codebook size. The cluster assignments $\mathbf{z}^i_{1:T}$ serve as the prediction targets of the model. Initially, $G$ can be an engineered image feature extractor such as Histogram of Oriented Gradients (HoG), analogous to MFCC in audio HuBERT. The intermediate layers of the HuBERT model are used as $G$ in later iterations.

To perform the masked prediction task, the model first encodes $\mathbf{I}_{1:T}$ using a ResNet into an intermediate visual feature sequence $\mathbf{f}^v_{1:T}$, which is then corrupted into $\tilde{\mathbf{f}}^v_{1:T}$ via a binary mask $M$. Specifically, $\forall t \in M, \tilde{\mathbf{f}}^v_t$ is replaced with a learned masked embedding. We adopt the same strategy in HuBERT to generate span masks. The masked visual features $\tilde{\mathbf{f}}^v_{1:T}$ are encoded into a sequence of contextualized features $\mathbf{e}_{1:T}$ via a transformer encoder followed by a linear projection layer. The loss is computed over the masked regions and optionally over unmasked ones (when $\alpha \geq 0$):

$$\mathbf{p}_t = \text{Softmax}(\mathbf{W}\mathbf{e}_t + \mathbf{b}), 1 \leq t \leq T$$
$$L = -\sum_{t \in M} \log p_t(z^i_t) - \alpha \sum_{t \notin M} \log p_t(z^i_t) \tag{1}$$

Where $(\mathbf{W} \in \mathbb{R}^{d \times V}, \mathbf{b} \in \mathbb{R}^V)$ are parameters of the projection layer which maps features into logits predicting the cluster assignments.

**Cross-modal Visual HuBERT:** The single-modal visual HuBERT aims to learn visual speech representation through gradually refined image features. However, it does not employ the audio stream of the video. Presumably, audio features, e.g., MFCC or a pre-trained audio HuBERT model, correlate with phones better than vanilla image features (e.g., HoG) do. To this end, we train an audio encoder based on the aligned audio frame sequence $\mathbf{A}_{1:T}$ in parallel to the visual encoder. The iterative training alternates between the two encoders. In each iteration, an audio encoder $E^a$ is utilized to generate target cluster assignments $\mathbf{z}^a_{1:T}$. The visual encoder $E^v$ is trained subsequently with $(\mathbf{I}_{1:T}, \mathbf{z}^a_{1:T})$. The $\mathbf{z}^a_{1:T}$ is also used to train the next iteration of the audio encoder $E^a$ for refinement.

The cross-modality visual HuBERT can be seen as modeling visual inputs by distilling knowledge from the audio stream, where $\mathbf{z}^a_{1:T}$ represents the audio-side knowledge. We hypothesize that the audio feature is more favorable to speech representation learning than the visual feature, which is validated in the Section E.1. Critical for the lip-reading downstream task, the masked prediction objective used by HuBERT forces the model to capture temporal relationships, which facilitates prediction of homophemes, which are groups of sounds with identical visual shapes (e.g., 'p'-'b', 'f'-'v', 'sh'-'ch') that are impossible to distinguish using a single image frame.

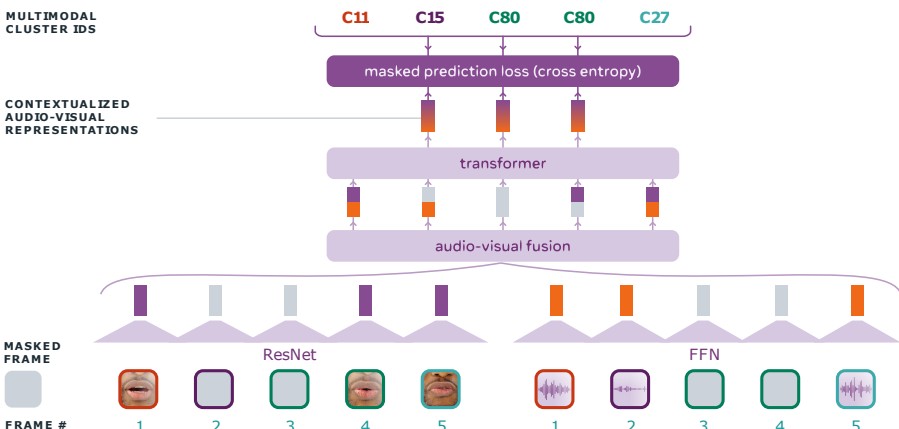

Figure 1: Illustration of AV-HuBERT. Masked prediction losses are only computed for the three middle frames, because at least one modality is masked for those frames. See section A for its comparison between single-modal and cross-modal visual HuBERT.

## 3.3 AUDIO-VISUAL HUBERT

Our primary model in this work is Audio-Visual HuBERT (AV-HuBERT), shown in figure 1, which is trained iteratively by alternating between feature clustering and masked prediction in a similar way to the Visual HuBERT but with four main improvements:

**Audio-visual input:** The AV-HuBERT model consumes both acoustic and image frames for the masked prediction training, which enables better modeling and distillation of the correlations between the two modalities. Specifically, image sequences and acoustic features pass through their light-weight modality-specific encoders to produce intermediate features, which are then fused and fed into a shared backbone transformer encoder to predict masked cluster assignments. The targets are generated from clustering audio features or features extracted from the previous iteration of the AV-HuBERT model. When fine-tuned for lip-reading, we drop the audio input to work solely with the visual input. The input discrepancy is addressed by modality dropout described next.

**Modality dropout:** Audio-visual speech recognition models can relate audio input to lexical output more effortlessly than the visual input stream, as observed in the literature (Afouras et al., 2018a; Ma et al., 2021b). This causes the audio modality to dominate model decisions. The problem is aggravated in our setting because the target cluster assignments are initially generated from acoustic features. To prevent the model's over-reliance on the audio stream in our joint model, we only use a linear layer to encode acoustic input to force the audio encoder to learn simple features.

Additionally, before fusing audio and visual inputs into the backbone transformer encoder, dropout is applied to mask the full features of one modality; we refer to it as modality dropout. With a probability $p_m$, both modalities are used as input. When only one modality is used, the audio stream is selected with a probability of $p_a$. Formally, given the encoded audio and visual feature sequence $\mathbf{f}_{1:T}^a$ and $\mathbf{f}_{1:T}^v$, equation 2 shows feature fusion equipped with modality dropout:

$$\mathbf{f}_t^{av} = \begin{cases} \texttt{concat}(\mathbf{f}_t^a, \mathbf{f}_t^v) & \text{with } p_m \\ \texttt{concat}(\mathbf{f}_t^a, \mathbf{0}) & \text{with } (1-p_m)p_a \\ \texttt{concat}(\mathbf{0}, \mathbf{f}_t^v) & \text{with } (1-p_m)(1-p_a) \end{cases} \tag{2}$$

where `concat` denotes channel-wise concatenation. Note that modality drop out is applied at the sequence level instead of at the frame-level, which effectively tasks AV-HuBERT to perform masked prediction with visual-only, audio-only, or audio-visual input. Modality dropout prevents the model from ignoring video input and encourages the model to produce the prediction regardless of what modalities are used as input. Furthermore, since the fine-tuning and inference phases use the visual stream alone (no audio input), this modality dropout mechanism bridges the gap between pre-training (multimodal) and fine-tuning/inference (single-modality). A similar dropout mechanism is used in prior work (Zhang et al., 2019a; Makino et al., 2019; Neverova et al., 2014; Abdelaziz et al., 2020) to increase the robustness in multi-modal settings. We verify the modality dropout effectiveness in Section D.

**Audio-visual clustering:** One benefit of pre-training on both modalities is the ability to generate multimodal cluster assignments that serve as target labels for the masked prediction task of the next iteration. In contrast to the Cross-modal Visual HuBERT where targets are generated from audio-based features or a prior Audio HuBERT model, the targets for AV-HuBERT are naturally multimodal after the first iteration. Lip movement sequences provide complementary information to the audio stream. Combining both modalities produces cluster assignments of higher quality for AV-HuBERT, as shown in Section E.1.

**Masking by substitution:** We propose a novel masking strategy for AV-HuBERT that masks segments in the visual stream by substituting them with random segments from the same video. More formally, given an input video $\mathbf{I}_{1:T}^v$, an imposter video $\mathbf{I}_{1:T_f}^{v,f}$ and a mask consisting of $n$ intervals $M = \{(s_i, t_i)\}_{1 \leq i \leq n}$, we corrupted $\mathbf{I}_{1:T}^v$ into $\tilde{\mathbf{I}}_{1:T}^v$ by setting:

$$\tilde{\mathbf{I}}_{s_i:t_i}^v = \mathbf{I}_{p_i:p_i+t_i-s_i}^{v,f}, \forall 1 \leq i \leq n \tag{3}$$

where $p_i$ is a sampled integer offset from the interval $[0, T_f - t_i + s_i]$. Now, to solve the task, the model needs to *first identify the fake frames* and then *infer the labels belonging to the original frames*. Since the "filled-in" segments are from real video segments and temporally smooth, the fake segment detection sub-task becomes less trivial compared to when using vanilla masking or substitution with non-consecutive frames. We show an ablation confirming the advantage of the proposed masking strategy in Section D.

The audio and visual segments are masked independently using two different masking probabilities $m_a$ and $m_v$. We hypothesize that the difficulty of the masked prediction task differs for each modality: inferring the masked targets given the audio stream is more straightforward than using the lip movement stream. Setting a high masking probability for acoustic frames is essential to help the whole model capture the language characteristics. On the contrary, setting a high masking probability for the visual input provide the model with more imposter segments than the original ones, hurting its ability to learn meaningful features (studied in Section D of the appendix). Given the output probability $\mathbf{p}_{1:T}$ and target cluster assignments $\mathbf{z}_{1:T}$, the AV-HuBERT pre-training loss is:

$$L = -\sum_{t \in M^a \cup M^v} \log p_t(z_t) - \alpha \sum_{t \notin M^a \cup M^v} \log p_t(z_t) \tag{4}$$

where $M^a$ and $M^v$ denotes the frames that are masked for the audio and the visual stream. $\alpha$ is a hyperparameter weighting the contribution of the unmasked regions in the overall objective.

### 3.4 Fine-tuning

The proposed pre-training approaches can be fine-tuned for visual speech recognition using any sequence classification loss. In this paper, we focus on fine-tuning with the connectionist temporal classification (CTC) (Graves et al., 2006) and attention-based sequence-to-sequence cross-entropy loss (Bahdanau et al., 2016) (S2S for brevity), which are the most popular choices. Assume that the feature sequence output of our pre-trained model is $\mathbf{e}_{1:T}$ and the ground-truth transcription is $w = w_1, w_2, ..., w_s$. For CTC, a projection layer is used to map the visual feature sequence into the output probabilities: $\mathbf{p}_t = \text{Softmax}(\mathbf{W}^{ft}\mathbf{e}_t + \mathbf{b}^{ft})$, where $\mathbf{W}^{ft} \in \mathbb{R}^{d \times (U+1)}$, $\mathbf{b}^{ft} \in \mathbb{R}^{U+1}$ and $U$ is the output vocabulary size (+1: plus blank symbol). The model is trained with CTC loss: $L_{ctc} = -\log \sum_{\pi \in \mathcal{B}^{-1}(w)} p(\pi|\mathbf{e}_{1:T})$, where $\mathcal{B}$ maps an alignment sequence $\pi$ to $w$. For S2S, a tranformer decoder is appended to the pre-trained encoder to autoregressively decode the feature sequence $\mathbf{e}_{1:T}$ into target probabilities $p(w_t|w_{1:t-1}, \mathbf{e}_{1:T})$. The whole model is trained with cross entropy loss: $L_{s2s} = -\sum_{t=1}^s \log p(w_t|w_{1:t-1}, \mathbf{e}_{1:T})$.

## 4 Experiment

### 4.1 Setup

We conduct experiments on two datasets: LRS3 (Afouras et al., 2018b) with 433 hours of transcribed English videos and VoxCeleb2 (Chung et al., 2018) with 2442 hours of unlabeled multilingual videos. We only use the English portion of VoxCeleb2, which amounts to 1,326 hours

of content. The inputs to our backbone model are lip Regions-Of-Interest (ROIs) for the visual stream and log filterbank energy feature for the audio stream. The image encoder is a modified ResNet-18, which has been used in prior work (Ma et al., 2021b; Martinez et al., 2020; Stafylakis & Tzimiropoulos, 2017). The audio encoder is simply a linear projection layer. We consider two model configurations: BASE with 12 transformer blocks and LARGE with 24 transformer blocks. For BASE and LARGE, the embedding dimension/feed-forward dimension/attention heads in each transformer block are 768/3072/12 and 1024/4096/16 respectively. The number of parameters in BASE and LARGE are 103M and 325M respectively. $\alpha$ in equation 4 is set to 0.

The model uses five iterations of feature clustering and masked prediction during pre-training. See Section B.4, E for details on clustering. For fine-tuning, we use phone targets for the CTC loss and unigram-based subword units (Kudo, 2018) for the S2S loss. For decoding the CTC-trained model, we use a 4-gram language model trained on LRS3 training text. For the S2S fine-tuned model, we rely only on its own decoder module to incorporate language information, with no external language model employed during inference. To further show the complementary relationship between AV-HuBERT and existing approaches of using unlabeled data, we also experiment on combining AV-HuBERT with self-training. Specifically, we generate pseudo labels for unlabeled data using a fine-tuned HuBERT, and fine-tune the pre-trained AV-HuBERT model with the combination of pseudo-labeled videos and original labeled videos. Note that no additional data is used when combined with self-training. More details about the used datasets, data pre-processing, and model training are in Section B.

## 4.2 MAIN RESULT

Table 1 compares the performance of our AV-HuBERT pre-training approach to previously published supervised, semi-supervised, and self-supervised lip-reading systems using different amounts of labeled and unlabeled data. Since the CTC and S2S fine-tuning approaches have similar trends, only S2S results are shown in table 1. Complete results of CTC fine-tuning are in Table C.4.[1]

Using 1,759 hours unlabeled data for pre-training and only 30 hours of labeled data for fine-tuning, AV-HuBERT-LARGE outperforms all the prior lip-reading models, including the model in (Makino et al., 2019) which is trained with 1000 times more labeled data. Fine-tuning with the whole training set of LRS3 further reduces WER. Combining our method and self-training achieves a new SOTA result with only 7% of the data used for training the model in Makino et al. (2019). Furthermore, it shows that AV-HuBERT and self-training are complementary to each other. Note the overall gain is attributed mainly to AV-HuBERT as self-training alone leads to much worse performance ($> 50\%$ WER). More details can be found in section C.3. Many prior models pre-train their visual front-end, e.g., ResNet-18, using word-level annotated lip-reading videos, which is costly to collect since they require word boundary information. In contrast to these models, our models are fully pre-trained from scratch using the proposed approach.

Compared to the semi-supervised approach Jasper-KD (Afouras et al., 2020), which transcribed 334 hours of the English data in VoxCeleb2 using a pre-trained ASR system,[2] our best model achieves 29% lower absolute WER benefiting from VoxCeleb2 for pre-training. Even when limiting our model to the LRS3 data for pre-training and fine-tuning, our model surpasses their semi-supervised system by 18%. Compared to LiRA (Ma et al., 2021a), a recently proposed self-supervised model for lip-reading, AV-HuBERT-BASE provides 17.5% lower absolute WER on average for low-resource and high-resource settings. The implementation details of LiRA are provided in Section B.5.

With the same network architecture, our pre-training approach significantly reduces WER compared to training from scratch, in both low-resource ($92.3\% \rightarrow 32.5\%$, LARGE) and high-resource ($62.3\% \rightarrow 28.6\%$, LARGE) settings. A qualitative view of the improvement can be found in Section F. Additionally, we notice that our AV-HuBERT pre-training helps under a fully supervised setting. Using the LRS3 data only (433 hours), pre-training followed by fine-tuning (41.6%, LARGE) outperforms training a lip-reading model from scratch (62.3%, LARGE) to predict the output text.

---

[1]The prior work in (Ma et al., 2021b) uses an outdated version of LRS3 (before 2018) with speaker overlap in training and test data, which is no longer publicly available. Its best results on the current version are included in table 1& C.4 . For comparison, we simulate the old closed-speaker setting in Section C.2.

[2]The gap in data amount (334hr vs 1,759hr) is due to the different parameters and ASR model used for filtering non-English data in VoxCeleb2.

Table 1: WER (%) of our models and prior work on the LRS3 dataset. †We re-implemented Ma et al. (2021a) with the same architecture since the author source code was not provided.

| Method | Backbone | Criterion | Labeled iso (hrs) | Labeled utt (hrs) | Unlabeled data (hrs) | WER (%) |
|---|---|---|---|---|---|---|
| *Supervised* | | | | | | |
| Afouras et al. (2020) | CNN | CTC | 157 | 433 | - | 68.8 |
| Zhang et al. (2019b) | CNN | S2S | 157 | 698 | - | 60.1 |
| Afouras et al. (2018a) | Transformer | S2S | 157 | 1,362 | - | 58.9 |
| Xu et al. (2020) | RNN | S2S | 157 | 433 | - | 57.8 |
| Shillingford et al. (2019) | RNN | CTC | - | 3,886 | - | 55.1 |
| Ma et al. (2021b) | Conformer | CTC+S2S | - | 433 | - | 46.9 |
| Ma et al. (2021b) | Conformer | CTC+S2S | 157 | 433 | - | 43.3 |
| Makino et al. (2019) | RNN | Transducer | - | 31,000 | - | **33.6** |
| *Semi-Supervised & Self-Supervised* | | | | | | |
| Afouras et al. (2020) | CNN | CTC | 157 | 433 | 334 | 59.8 |
| Ma et al. (2021a)† | Transformer-BASE | S2S | - | 30 | 433 | 71.9 |
| | | | - | 433 | 1,759 | 49.6 |
| *Proposed (Self-Supervised & Self-Supervised + Semi-Supervised)* | | | | | | |
| AV-HuBERT | Transformer-BASE | S2S | - | 30 | - | 94.3 |
| | | | - | 30 | 433 | 51.8 |
| | | | - | 30 | 1,759 | **46.1** |
| | | | - | 433 | - | 60.3 |
| | | | - | 433 | 433 | 44.0 |
| | | | - | 433 | 1,759 | **34.8** |
| | Transformer-LARGE | S2S | - | 30 | - | 92.3 |
| | | | - | 30 | 433 | 44.8 |
| | | | - | 30 | 1,759 | **32.5** |
| | | | - | 433 | - | 62.3 |
| | | | - | 433 | 433 | 41.6 |
| | | | - | 433 | 1,759 | **28.6** |
| AV-HuBERT + Self-Training | Transformer-LARGE | S2S | - | 30 | 1,759 | **28.6** |
| | | | - | 433 | 1,759 | **26.9** |

AV-HuBERT, pre-trained on video-audio pairs, learns better fine-grained visual representation than the scratch model trained on video-text pairs. The benefits of AV-HuBERT pre-training in various labeled data setups can be found in Section C.1.

## 4.3 AV-HuBERT vs. Visual HuBERT

We compare AV-HuBERT against a suite of alternatives, including the Single-modal and Cross-modal Visual HuBERT in Table 2. All the models are BASE pretrained on 433 hours of unlabeled data and fine-tuned on 30 hours of labeled data. For this comparison, we use CTC fine-tuning due to its computational efficiency and given their similarity in results trends to S2S.

Table 2: Fine-tuning performance (in WER, %) of AV-HuBERT and visual HuBERT on different target labels. Init: feature in the initial iteration, sub: feature in subsequent iterations. AV: AV-HuBERT, V: Visual-HuBERT, A: Audio-HuBERT.

| Model/init→sub | Iteration | | | | |
|---|---|---|---|---|---|
| | 1 | 2 | 3 | 4 | 5 |
| AV/MFCC→AV | 71.5 | 63.6 | 60.9 | 58.8 | 58.2 |
| AV/MFCC→A | 71.5 | 64.3 | 63.5 | - | - |
| V/MFCC→A | 75.4 | 69.4 | 69.1 | - | - |
| V/MFCC→V | 75.4 | 72.6 | 72.3 | - | - |
| V/HoG→V | 80.3 | 80.1 | - | - | - |

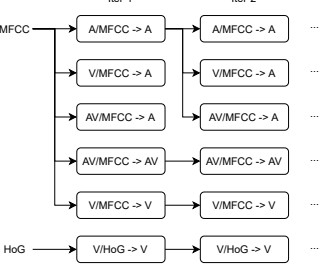

As shown in table 2, target cluster assignments driven from a single modality, either audio or visual, do not provide much WER reduction beyond the second iteration. Training the AV-HuBERT model

using targets driven from audio-visual features keeps improving for more iterations and achieves better final performance. As mentioned in Section 3, visual information is complementary to audio input; hence, using both produces higher quality target clusters. Measuring the quality of different target labels using cluster quality metrics such as purity and NMI shows the same trend observed from lip-reading WER (See appendix E.1).

Fixing target labels used for the masked-prediction pre-training, AV-HuBERT (AV/MFCC→A) outperforms the Cross-modal Visual HuBERT (V/MFCC→A) by a large margin. AV-HuBERT effectively transfers knowledge from audio into the visual encoder and the backbone transformer model to benefit visual-only fine-tuning and inference. In contrast to AV-HuBERT, iterative pre-training brings much smaller gains to single-modality visual HuBERT ("V/MFCC→V", "V/HoG→V").

Starting with audio features is critical for learning effective target labels for the masked-prediction task. Phonetic information, which is crucial for lip-reading, is primarily present in the audio stream. All the models considered so far are based on audio feature MFCC clustering in their initial iteration. As is shown in the "V/HoG→V" row, using hand-engineered visual features provides a lousy starting point for iterative learning. Visual features such as HoG mainly incorporate low-level visual cues such as edges and luminance, which is irrelevant to the downstream recognition task. Using features of a higher correlation with phonetic units are more likely to benefit the final lip-reading model. Indeed, clustering MFCC features show much higher purity (30.3%) than HoG clusters (16.4%) if one considers phone labels as the target units, as shown in Table E.1.

## 4.4 MULTILINGUAL VS. MONOLINGUAL

Since the correlation between lip movements and the produced sound is governed by the vocal apparatus that is language-agnostic, the proposed AV-HuBERT can utilize multi-lingual data for such learning. This would be particularly beneficial for low-resource languages. Nevertheless, the masked language modeling aspect in AV-HuBERT is still language-dependent, implying that mixing other languages would increase the language domain discrepancy. To study how these two factors affect AV-HuBERT, we compare using monolingual English data only versus multilingual videos in the pre-training phase. In particular, we vary the amount of English data we use in pre-training while the number of non-English utterances is fixed to 1,116 hours in all the settings. For simplicity, we train AV-HuBERT (BASE) for one iteration with MFCC clusters and fine-tune it with CTC using 30 hours of labeled data.

Table 3: WER (%) with different amounts of unlabeled English utterances in pre-training. Non-En data: 1,116 hours. Labeled data for fine-tuning: 30 hours.

| Hours of En data for Pre-Training | 0 | 100 | 400 | 800 | 1759 |
|---|---|---|---|---|---|
| Pre-train on En only, WER (%) | 84.1 | 77.8 | 68.9 | 67.9 | **59.9** |
| Pre-train on En + 1,116 hr of non-En, WER (%) | **70.6** | **68.4** | **67.4** | **66.6** | 64.3 |

As is shown in table 3, using non-English data in pre-training significantly reduces WER when there are no or very little English data in pre-training ($\leq$ 100 hours). As we increase the amount of English data, the gain diminishes because the out-of-domain effect brought by non-English data outweighs the benefit of the overall increase in pre-training data. Using the whole English data only for pre-training is better than combining it with other languages (59.9% vs. 64.3%). Training with 5 iterations leads to similar results (47.3% vs. 48.9%). This experiment highlights the importance of the domain match between pre-training and fine-tuning data. For zero/low-resource settings, merging data from other languages in pre-training benefits the downstream task. When the unlabeled data from the target language is abundant, limiting the pre-training data to one language is beneficial.

## 4.5 ASR PERFORMANCE

The multimodal clusters produced by AV-HuBERT, which have higher quality than audio-HuBERT targets, can also benefit speech pre-training. To test our hypothesis, we trained an *audio-HuBERT*, with only audio input during the masked-prediction pre-training, for one iteration with cluster assignments driven from AV-HuBERT features. We also pre-trained an audio-HuBERT from scratch

using clusters driven from the MFCC features for three iterations. The two pre-trained models are evaluated on a downstream ASR task.

Table 4 shows the performance of different models fine-tuned on the ASR task. We only include the performance of S2S fine-tuning for our models as it consistently outperforms the CTC fine-tuning. An audio-HuBERT pre-trained using targets generated by AV-HuBERT features outperforms the vanilla audio-HuBERT in low-resource (30h) and high-resource settings (433h) fine-tuning settings across different model architectures. With the same amount of labeled data, our best model (1.4%) outperforms the prior SOTA (2.3%) even without an external language model during inference.

Given that the AV-HuBERT model utilizes both modalities at its input, it can be fine-tuned, in principle, for the ASR downstream task. In practice, we notice pre-training an audio-HuBERT with audio-visual cluster leads to better ASR performance (3.8%) than a pre-trained AV-HuBERT (4.6%), potentially due to its hyperparameters being selected based on lip reading rather than ASR. In fact, audio-HuBERT can be treated as a special case of AV-HuBERT with $p_m = 0$, $p_a = 1$.

Table 4: ASR WER (%) of audio-HuBERT pre-trained with audio-only/audio-visual clusters and their comparison to prior work on the LRS3 dataset.

| Method | Backbone | Criterion | LM | Labeled data (hrs) | Unlabeled data (hrs) | WER (%) |
|---|---|---|---|---|---|---|
| *Supervised* | | | | | | |
| Afouras et al. (2018a) | Transformer | S2S | √ | 1,362 | - | 8.3 |
| Afouras et al. (2018a) | Transformer | CTC | √ | 1,362 | - | 8.9 |
| Xu et al. (2020) | RNN | S2S | - | 433 | - | 7.2 |
| Ma et al. (2021b) | Conformer | CTC+S2S | √ | 433 | - | **2.3** |
| *Self-Supervised* | | | | | | |
| | | | - | 30 | 433 | 5.4 |
| | Transformer-Base | S2S | - | 30 | 1,759 | 5.0 |
| Hsu et al. (2021a) | | | - | 433 | 1,759 | 2.4 |
| (A/MFCC→A) | | | - | 30 | 433 | 4.5 |
| | Transformer-Large | S2S | - | 30 | 1,759 | 3.2 |
| | | | - | 433 | 1,759 | **1.5** |
| *Proposed (Self-Supervised)* | | | | | | |
| | | | - | 30 | 433 | 4.9 |
| | Transformer-Base | S2S | - | 30 | 1,759 | 3.8 |
| A/MFCC→AV | | | - | 433 | 1,759 | 2.0 |
| | | | - | 30 | 433 | 4.2 |
| | Transformer-Large | S2S | - | 30 | 1,759 | 2.9 |
| | | | - | 433 | 1,759 | **1.3** |

## 5 CONCLUSION

We presented multiple pre-training models for visual speech recognition. Our AV-HuBERT model leverages the strong correlation between the audio and lip movement streams for self-supervised audio-visual speech representation learning. Our pre-training approaches iteratively alternate between feature clustering and learning new features through a masked-prediction loss. The AV-HuBERT model consumes masked image and audio frames to predict target cluster assignments. The targets are initially generated from MFCC features and gradually refined through iterative training. Experiments on visual speech recognition show that AV-HuBERT achieves SOTA using 433 hours of text transcriptions, two orders of magnitude less than the 31,000 hours of labeled data used in the prior best approach. When using only one-thousandth of labeled data, the lip-reading performance outperforms the prior SOTA by more than 10% (relative). AV-HuBERT also improves the representation for the ASR downstream task. An audio-HuBERT model trained with targets generated by an AV-HuBERT model shows superior performance, achieving the SOTA in the audio-based speech recognition in the LRS3 dataset. As future work, AV-HuBERT can be applied for multilingual lip-reading in low-resource languages. Additionally, our approach can be extended to other applications of visual speech representation, such as speech enhancement and generation.

ETHICAL STATEMENT

All the data used in this paper are publicly available and are used under the following three licenses: the TED terms of use, the Creative Commons BY-NC-ND 4.0 license and Creative Commons Attribution 4.0 International License. Through spot-checking, we find the datasets are gender balanced and cover a wide range of races and ages. However, the distribution of speakers in the data may not be representative of the global human population. Please be cautious of unintended societal, gender, racial and other biases caused by the fact. To maintain anonymity, only the mouth area of a speaker is visualized wherever used in the paper. The proposed method can be applied in several areas including security and crime investigations. However it can also be used for malicious purposes such as surveillance and wiretapping. We are committed to distributing our code and model carefully, with special attention to any potential security and privacy concerns.

REPRODUCIBILITY STATEMENT

Our code and models are publicly available. In the meantime, we include as many implementation details as we can in the paper.

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

# A MODEL ILLUSTRATION

Figure A.1: Comparison between the proposed AV-HuBERT with single-modal and cross-modal visual HuBERT

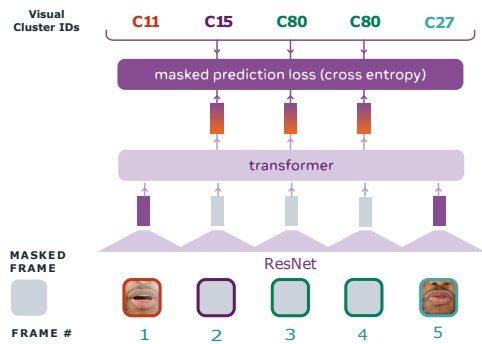

(a) Single-modal visual HuBERT

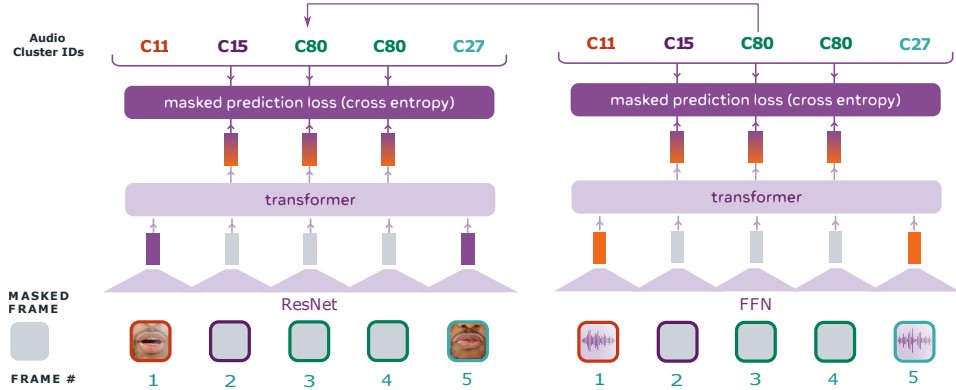

(b) Cross-modal visual HuBERT

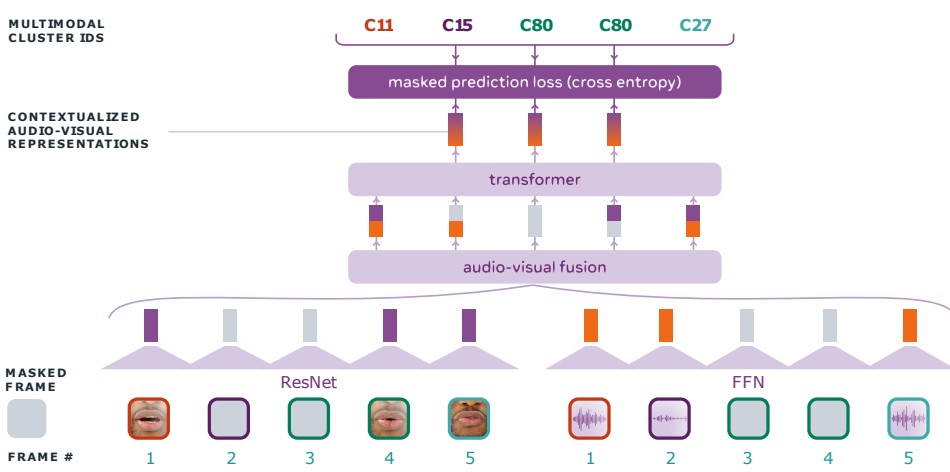

(c) Audio-visual HuBERT (proposed)

# B DETAILED EXPERIMENTAL SETUP

## B.1 DATASETS

**LRS3** (Afouras et al., 2018b) is the largest publicly available sentence-level lip reading dataset to date. It consists of over 400 hours of video, extracted from TED & TEDx talks in English from YouTube. In the original dataset, the training data is split into two partitions: pretrain (403 hours) and trainval (30 hours). Both parts are transcribed at the sentence level and come from the same source as the test set. The pretrain set differs from trainval in that the duration of its video clips are of a much wider range and can be shorter or longer than a full sentence. In the low-resource setup, we only use trainval as the labeled data. As no official development set is provided, we randomly select 1,200 sequences from trainval as the validation set (about 1 hour) for early stopping and hyperparameter tuning.

**VoxCeleb2** (Chung et al., 2018) is originally created for multilingual audio-visual speaker recognition and it contains over 2,442 hours of utterances of over 6,000 speakers extracted from YouTube videos. The dataset naturally serves our purpose as it does not contain ground-truth transcriptions. The VoxCeleb2 has substantial domain discrepency to the LRS3 data as its utterances are from multiple languages and includes videos in a larger variety of domains including interviews, talks, excerpts under indoor and outdoor environments. In VoxCeleb2, by default we only use the English portion for pre-training. As no ground-truth language label is given in the VoxCeleb2, we use a simple heuristic to choose the English samples. Specifically, we use an off-the-shelf character-based ASR model (Hsu et al., 2021a) trained on Librispeech which achieves 1.9%/3.5% WER on clean/other test set. We run greedy decoding on the VoxCeleb2 and use the proportion of valid English words to determine if a target utterance is English or not. An utterance is only selected if the proportion of valid English words is higher than 60%. The total amount of unlabeled data after filtering is 1,326 hours.

## B.2 DATA PREPROCESSING

For each video clip, we detect the 68 facial keypoints using dlib (King, 2009) and align each frame to a reference face frame via affine transformation. We crop a $96 \times 96$ region-of-interest (ROI) centered on the mouth. Each image frame is converted to grayscale. We randomly cropped $88 \times 88$ from the whole ROI and randomly flipped it horizontally with probablity of 0.5 during training. At test time, $88 \times 88$ ROI is center cropped and does not go through horizontal flipping. The preprocessing steps remain same as prior works in lip reading (Martinez et al., 2020; Ma et al., 2021b). For the associated audio, we extract the 26-dimensional log filterbank energy feature at a stride of 10 ms from the raw waveform and use it as input to the model. As the image frames are sampled at 25Hz, we stack the 4 neighboring acoustic frames to synchronize the two modalities.

## B.3 AV-HUBERT MODEL ARCHITECTURE

In the modified ResNet-18 (Ma et al., 2021b; Martinez et al., 2020; Stafylakis & Tzimiropoulos, 2017), the first convolutional layer is substituted by a 3D convolutional layer with kernel size $5 \times 7 \times 7$. The visual feature tensor is flattened into a single-dimensional vector through a 2D average pooling layer in the end. We use one linear projection layer as the audio encoding module. The acoustic features are normalized by per-frame statistics before feeding into the network (Ba et al., 2016). We use a dropout of $p = 0.1$ after the self-attention block within each transformer layer, and each transformer layer is dropped (Fan et al., 2020) at a rate of 0.1.

## B.4 TRAINING AND INFERENCE

**Pretraining** Our models are implemented with fairseq (Ott et al., 2019). The whole network is randomly initialized before pre-training. In pre-training, the model is trained for five iterations in total. For the initial iteration, we generate the targets by running k-means clustering algorithm on 39-dimensional MFCC feature (13 coefficients with its first- and second-order derivatives) extracted from raw audio waveform. For the subsequent iterations, the feature from an intermediate layer of the model in the previous iteration is used for clustering. The layer index (one-based) used for clustering in iteration 1-4 are {9, 12, 12, 12}. The number of features are clustered to {100, 100,

500, 1000, 2000} respectively for the 5 iterations. See section E.3 for analysis. To save training time, we always use the BASE model to generate clusters and LARGE model is only trained in the 5th iteration.

We set both $p_m$ and $p_a$ to 0.5 for modality dropout at training time. To extract features for clustering, both modalities are used. We adopt the strategy used in wav2vec 2.0 (Baevski et al., 2020) to generate masks, where $p\%$ of all frames are randomly selected as start and subsequent $l$ frames are masked. In iteration 1-4, we mask the fused features and set $p/l$ to be 8/10 respectively as we observe such practice generates higher quality cluster assignments (see section E.2). In the last iteration, we set $p/l$ to be 6/5 for video and 8/10 for audio (see section D).

We train the model with Adam (Kingma & Ba, 2015), warming up the learning rate for the first 8% of updates to a peak of 0.002 and then linearly decay it. Videos are batched together to not exceed 1,000 image frames (40 seconds) per GPU. Both BASE and LARGE models are updated for 400K and 600K steps at each iteration, respectively in 433h/1759h unlabeled settings. We train on 32 and 64 V100-GPUs for BASE and LARGE. On average, each iteration takes $\sim 2.0/3.0$ days for BASE and $\sim 2.4/3.6$ days for LARGE in using 433h/1759h unlabeled data for pre-training.

**Fine-tuning** After pre-training, we fine-tune the AV-HuBERT model on labeled (video, text) pairs. The audio encoder is removed and its output is replaced by a zero-vector. In CTC fine-tuning, a randomly initialized projection layer is added on top of the transformer to map features into phonemes. The lexicon is constructed with CMUDict (cmu). In fine-tuning with S2S, we use a 6-layer/9-layer transformer decoder on BASE and LARGE model to decode features into unigram-based subword units (Kudo, 2018). The vocabulary size in CTC and S2S are 46 and 1000 respectively.

In CTC, the pre-trained model is updated from the initial iteration without any freezing. The model is fine-tuned for 30K/100K steps respectively in 30h/433h setting. In S2S, the pre-trained model (i.e., encoder) is frozen for the first $N\%$ updates . $N$ is 100 and 50 for 30h and 433h labeled setting respectively. The entire model is trained for 18K/45K steps in the 30h/433h setting. Both models are trained with Adam, with the learning rate being warmed up for the first $P\%$ of updates to a peak of 0.001 and linearly decayed. P is tuned among {10, 30, 50}. All hyperparamters are tuned on the validation set.

**Decoding** For CTC, we use a 4-gram language model trained on text data in the LRS3 training set. The perplexity of the 4-gram LM on test set is 110.5. No language model is used for S2S decoding. For CTC, we tune the beam width among {5, 10, 20, 50, 100, 150}, the language model weight among {0, 1, 2, 4, 8} and word insertion penalty among {$\pm4, \pm2, \pm1, 0$}. For S2S, The beam width and length penalty are tuned among {5, 10, 20, 50} and {$0, \pm1$}. The tuning is done on the validation set.

**Self-Training** We apply self-training on LARGE AV-HuBERT. Specifically, the fine-tuned LARGE HuBERT model(A/MFCC$\rightarrow$AV, Table 4) is used to assign pseudo-labels to the unlabeled audio-visual data. In 30h/433h setting, the amount of data for fine-tuning A/MFCC$\rightarrow$AV are 30h and 433h respectively. The pre-trained AV-HuBERT LARGE is fine-tuned with the pseudo-labeled videos and videos with ground-truth text labels (30h/433h). Note the data used here is exactly same with the case of using AV-HuBERT only.

### B.5  LiRA Implementation

We re-implemented the LiRA (Ma et al., 2021a) training objective in our framework, as there does not exist publicly available implementations and we aim to focus the comparison on the pre-training objective rather than the architectural difference. We use the same backbone architecture as the BASE AV-Hubert except the output layer being a linear project layer with an output dimension of 256. The 256-dimensional frame PASE+ feature is extracted from the audio with its official implementation in (Ravanelli et al., 2020). The original PASE+ feature is downsampled to 25Hz for synchronization with the visual stream. The pre-trained model is fine-tuned in both CTC and S2S. The optimizer and learning rate schedule in pre-training, hyperparameter search in fine-tuning and decoding remain the same as AV-HUBERT. Note with our implementation, the WER is reduced by 23% ($94.3\% \rightarrow 71.9\%$) using LiRA when the percentage of labeled data is 6.9% (30h labeled, 433h in total), while Ma et al. (2021a) achieves $\sim 10\%$ improvement in a similar setting.

# C    ADDITIONAL LIP-READING RESULTS

## C.1    AMOUNT OF LABELED DATA

Table C.1 shows the effect of pre-training on different amount of labeled data for fine-tuning. We use 433 hours of unlabeled data (LRS3 only) and randomly selected 1, 10 and 100 hours of labeled data for fine-tuning. Overall pre-training brings large and consistent gains across different amount of labeled data. Specifically CTC-based fine-tuning outperforms S2S-based fine-tuning in low-resource settings (1-hour and 10-hour). The larger number of parameters as well as the lack of a language model for decoding makes S2S model more likely to overfit especially when the amount of fine-tuning data is small.

Table C.1: WER (%) in using different amount of labeled data for fine-tuning (BASE, 433 hours unlabeled)

| Labeled (hrs) | Unlabeled (hrs) | Criterion | LM | WER (%) w/o pretrain | w/ pretrain |
|---|---|---|---|---|---|
| 1 | 433 | CTC | 4-gram | 98.6 | **68.8** |
|   |   | S2S | - | 98.9 | **92.0** |
| 10 | 433 | CTC | 4-gram | 90.8 | **57.6** |
|   |   | S2S | - | 97.6 | **63.1** |
| 100 | 433 | CTC | 4-gram | 77.8 | **54.2** |
|   |   | S2S | - | 84.3 | **48.1** |

## C.2    PERFORMANCE ON SEEN SPEAKERS

The current LRS3 benchmark is under the open-speaker setting, where the speaker identities in training and test set do not overlap. To test the lip reading performance for a fixed set of speakers which is the case for an early versions of LRS3 used before October 2018, we randomly choose 5 groups of utterances from the trainval partition of LRS3 as test set and repeat experiments for each group independently. Each group contains 1322 utterance, which is of the same amount as the original test set. The model we compare is the AV-HUBERT LARGE pre-trained with 1,759 unlabeled data. As is shown in table C.2, the average WER achieved by our model for seen speakers is $18.0 \pm 0.5\%$, which is significantly lower than the WER for unseen speakers (30.5%) under the open-speaker setting.

Table C.2: WER (%) under closed-speaker setting for 5 randomly sampled test sets and their average

|  | Test set (seen speakers) | | | | | AVG |
|---|---|---|---|---|---|---|
|  | 1 | 2 | 3 | 4 | 5 |  |
| WER (%) | 17.4 | 18.6 | 18.5 | 17.5 | 18.3 | $18.0 \pm 0.5$ |

## C.3    PERFORMANCE OF SELF-TRAINING ONLY

Table C.3 shows the performance of only applying self-training. The WER of a self-training only model is significantly higher than AV-HuBERT and self-trained AV-HuBERT, which suggests that the gain of the combined approach is primarily from AV-HuBERT.

## C.4    FULL RESULTS WITH CTC FINE-TUNING

Table C.4 shows the full results on LRS3, which includes the CTC fine-tuning performance for all the models we implemented. In general, the conclusions we draw from S2S (e.g., the benefits of our pre-training approach in different settings, the improvement over LiRA) in section 4.2 holds for CTC as well.

Table C.3: Comparison of WER (%) among model trained from scratch, self-training only, AV-HuBERT only and self-trained AV-HuBERT. All models are Transformer-LARGE.

| Labeled (hrs) | Unlabeled (hrs) | Method | WER (%) |
|---|---|---|---|
| 30 | 1,759 | w/o pre-training | 92.3 |
| | | Self-training | 53.0 |
| | | AV-HuBERT | 32.5 |
| | | AV-HuBERT + Self-training | **28.6** |
| 433 | 1,759 | w/o pre-training | 62.3 |
| | | Self-training | 51.7 |
| | | AV-HuBERT | 28.6 |
| | | AV-HuBERT + Self-training | **26.9** |

Table C.4: WER (%) of our models and the comparison with prior works on LRS3-TED dataset. †We re-implemented Ma et al. (2021a) using the same model architecture as our approach to have a more fair comparison.

| Method | Backbone | Criterion | Labeled iso (hrs) | Labeled utt (hrs) | Unlabeled data (hrs) | WER (%) |
|---|---|---|---|---|---|---|
| *Supervised* | | | | | | |
| Afouras et al. (2020) | CNN | CTC | 157 | 433 | - | 68.8 |
| Zhang et al. (2019b) | CNN | S2S | 157 | 698 | - | 60.1 |
| Afouras et al. (2018a) | Transformer | S2S | 157 | 1,362 | - | 58.9 |
| Xu et al. (2020) | RNN | S2S | 157 | 433 | - | 57.8 |
| Shillingford et al. (2019) | RNN | CTC | - | 3,886 | - | 55.1 |
| Ma et al. (2021b) | Conformer | CTC+S2S | - | 433 | - | 46.9 |
| Ma et al. (2021b) | Conformer | CTC+S2S | 157 | 433 | - | 43.3 |
| Makino et al. (2019) | RNN | Transducer | - | 31,000 | - | **33.6** |
| *Semi-Supervised & Self-Supervised* | | | | | | |
| Afouras et al. (2020) | CNN | CTC | 157 | 433 | 334 | 59.8 |
| Ma et al. (2021a)† | Transformer-BASE | CTC | - | 30 | 433 | 72.8 |
| | | | - | 433 | 1,759 | 58.4 |
| | | S2S | - | 30 | 433 | 71.9 |
| | | | - | 433 | 1,759 | 49.6 |
| *Proposed (Self-Supervised & Self-Supervised + Semi-Supervised)* | | | | | | |
| AV-HuBERT | Transformer-BASE | CTC | - | 30 | - | 83.7 |
| | | | - | 30 | 433 | 55.3 |
| | | | - | 30 | 1,759 | **47.3** |
| | | | - | 433 | - | 62.5 |
| | | | - | 433 | 433 | 49.3 |
| | | | - | 433 | 1,759 | **43.0** |
| | | S2S | - | 30 | - | 94.3 |
| | | | - | 30 | 433 | 51.8 |
| | | | - | 30 | 1,759 | **46.1** |
| | | | - | 433 | - | 60.3 |
| | | | - | 433 | 433 | 44.0 |
| | | | - | 433 | 1,759 | **34.8** |
| | Transformer-LARGE | CTC | - | 30 | - | 92.2 |
| | | | - | 30 | 433 | 48.4 |
| | | | - | 30 | 1,759 | **40.7** |
| | | | - | 433 | - | 61.9 |
| | | | - | 433 | 433 | 44.3 |
| | | | - | 433 | 1,759 | **38.6** |
| | | S2S | - | 30 | - | 92.3 |
| | | | - | 30 | 433 | 44.8 |
| | | | - | 30 | 1,759 | **32.5** |
| | | | - | 433 | - | 62.3 |
| | | | - | 433 | 433 | 41.6 |
| | | | - | 433 | 1,759 | **28.6** |
| AV-HuBERT + Self-Training | Transformer-LARGE | S2S | - | 30 | 1,759 | **28.6** |
| | | | - | 433 | 1,759 | **26.9** |

# D ABLATION STUDIES

The ablation studies in this section are done in the last iteration of the AV-HuBERT, pre-trained with 433 hours of unlabeled data. The model is fine-tuned with 30 hours of labeled data using CTC.

Table D.1: Ablation study for hyper-parameters. The ablations are done in the last iteration of AV-HUBERT. $m_a/m_v$: the probability of an acoustic/image frame being masked.

| | Masking | | | Modality Dropout | | Loss | WER | |
|---|---|---|---|---|---|---|---|---|
| Where | How | $m_a$ | $m_v$ | $p_m$ | $p_a$ | $\alpha$ | dev | test |
| Input | Sub (same, seg) | 0.8 | 0.3 | 0.5 | 0.5 | 0.0 | 46.8 | 55.3 |
| | Sub (same, frm) | | | | | | 47.2 | 55.8 |
| | Sub (diff, seg) | | | | | | 47.6 | 56.1 |
| | Learned Embedding | | | | | | 52.6 | 57.8 |
| | Gauss. Noise | | | | | | 52.4 | 57.9 |
| Feature | Learned Embedding | | | | | | 55.2 | 58.2 |
| Input | Sub (same, seg) | 0.8 | 0.3 | 0.5 | 0.5 | 0.0 | 46.8 | 55.3 |
| | | 0.8 | 0.8 | | | | 59.3 | 61.6 |
| | | 0.3 | 0.3 | | | | 54.9 | 58.2 |
| Input | Sub (same, seg) | 0.8 | 0.3 | 0.5 | 0.5 | 0.0 | 46.8 | 55.3 |
| | | | | 1.0 | n/a | | 55.2 | 57.0 |
| Input | Sub (same, seg) | 0.8 | 0.3 | 0.5 | 0.5 | 0.0 | 46.8 | 55.3 |
| | | | | | | 1.0 | 46.7 | 55.7 |

**Masking Strategy** In the first part of table D.1, we compare the proposed masking strategy against several alternatives. Feature masking applies span mask at feature level and leads to the worst performance, which is due to the leakage of information to ResNet. Directly masking image sequence with random Gaussian noise or a learned embedding slightly improves the performance by preventing the prior issue. However, those artificial frames also corrupts the raw image sequence and enlarges the domain gap in videos between pre-training and fine-tuning. On the other hand, our proposed method achieves better performance. Specifically, using segments from the same utterance ("Sub, (same, seg)") as the imposter leads to the best result compared to sampling from a different utterance ("Sub, (diff, seg)") or sampling non-consecutive frames from the same utterance ("Sub, (same, frm)").[3] The filled-in fake images are visually similar to the raw images and substitution with a segment keeps the temporal smoothness making the replaced segment more realistic, which enforces the ResNet to encode more fine-grained visual details.

**Masking probability** We set two different masking probabilities for audio and visual stream in the last iteration. The probabilities of an acoustic frame and image frame being masked are 0.8 and 0.3 respectively. As is shown in the second part of table D.1, setting masks for audio and visual stream independently is essential because the optimal masking probability for audio and visual stream are different. Audio encoder tends to deteriorate into a simple acoustic feature extractor when mask length is small. On the other hand, long visual mask will lead to the model lacking context to distinguish between fake and real images.

**Modality dropout** The third part of table D.1 compares the model performance with and without modality dropout. Randomly dropping audio sequence prevents the model from over-relying on audio for masked prediction and helps the visual representation learning.

**Where to compute prediction loss** In the last part of table D.1, we compare the choice of masked prediction vs. prediction. The loss weight on unmasked region does not have a large impact on the fine-tuning performance. This is different from the findings in Audio HuBERT Hsu et al. (2021a), where masked prediction leads to much better performance. Given image frames as input, the prediction of cluster assignments, which are mostly determined by the accompanied audio stream, helps encode phonetic information into the visual representation. The task is much less trivial than

---

[3]To avoid using original frames for substitution, $p_i$ in equation 3 is selected from $[0, 2s_i - t_i] \cup [t_i, T_f - t_i + s_i]$ if the imposter is from the same sequence.

a single-modal model (audio-HuBERT), where setting non-zero weight on unmasked prediction can easily make the model deteriorate into an acoustic feature extractor. In addition, the high quality of targets in the last iteration also makes such prediction more helpful.

# E  ANALYSIS ON CLUSTERING

## E.1  MEASURING CLUSTERING QUALITY

For analysis, we use frame-level phonetic labels as the ground-truth and match its correlation between cluster assignments. The phonetic labels are obtained via forced alignment from a monophone based HMM-GMM ASR model trained on LRS3. In particular, we use clustering purity and Normalize Mutual Information (NMI) as the evaluation metrics. Table E.1 shows that (1) The quality of cluster assignments is consistent with fine-tuning performance across different models (2) Hand-engineered audio feature (MFCC, NMI: 21.5%) has much stronger correlation with phonetic labels than the visual feature (HoG, NMI: 1.6%) (3) Audio-visual clusters (NMI: 44.2%) are of better quality than pure audio-based clusters (NMI: 39.7%). (4) In single-modality visual Hubert (V/HoG→V), feature quality is improved negligibly through iteration training.

Table E.1: Quality of different cluster assignments. Each number is in the format of Purity (NMI). The metrics of cluster assignments and WER in last iteration of each model are in boldface.

| Model | Iter | Feature | Target K | Purity (%),↑ | NMI (%),↑ | WER (%),↓ |
|---|---|---|---|---|---|---|
| AV/MFCC→AV (Proposed) | 1 | MFCC | 100 | 30.3 | 21.5 | 71.5 |
| | 2 | AV/MFCC→AV (it1, L9) | 100 | 47.3 | 37.7 | 63.6 |
| | 3 | AV/MFCC→AV (it2, L12) | 500 | 61.5 | 42.6 | 60.9 |
| | 4 | AV/MFCC→AV (it3, L12) | 1000 | 65.6 | 43.7 | 58.8 |
| | 5 | AV/MFCC→AV (it4, L12) | 2000 | **68.8** | **44.2** | **58.2** |
| AV/MFCC→A | 1 | MFCC | 100 | 30.3 | 21.5 | 71.5 |
| | 2 | A/MFCC→A (it1, L9) | 100 | 47.0 | 36.7 | 64.3 |
| | 3 | A/MFCC→A (it2, L12) | 500 | **56.5** | **39.7** | **63.5** |
| V/MFCC→A | 1 | MFCC | 100 | 30.3 | 21.5 | 75.4 |
| | 2 | A/MFCC→A (it1, L9) | 100 | 47.0 | 36.7 | 69.4 |
| | 3 | A/MFCC→A (it2, L12) | 500 | **56.5** | **39.7** | **69.1** |
| V/MFCC→V | 1 | MFCC | 100 | 30.3 | 21.5 | 75.4 |
| | 2 | V/MFCC→V (it1, L9) | 100 | 32.8 | 22.9 | 72.6 |
| | 3 | V/MFCC→V (it2, L12) | 500 | **33.0** | **22.8** | **72.3** |
| V/HoG→V | 1 | HoG | 100 | 16.4 | 1.6 | 80.3 |
| | 2 | V/HoG→V (it1, L9) | 100 | **16.4** | **1.8** | **80.1** |

## E.2  FEATURE MASKING PRODUCES BETTER FEATURES FOR CLUSTERING

In iteration 1-4, we apply the mask in the fused feature. We observe such practice generates targets of higher quality, thus helping future iterations more. Table E.2 shows a comparison between such two different masking strategies. Input-level masking enhances the learning of visual representation (see table D.1) while produces worse audio-visual feature (NMI: 27.2%). In contrast, the two streams of original input are better aligned in feature-level masking which is consistent with the cluster generation process, thus leading to better audio-visual clusters (NMI: 37.7%).

Table E.2: Impact of masking strategy on quality of cluster assignments (purity/NMI: quality of cluster assignments used to train the model)

| Feature | K | Purity (%) | NMI (%) |
|---|---|---|---|
| MFCC | 100 | 30.3 | 21.5 |
| AV/MFCC→AV (it1, L9) w/ Feature Masking | 100 | 47.3 | 37.7 |
| AV/MFCC→AV (it1, L9) w/ Input Masking | 100 | 34.5 | 27.2 |

## E.3  CLUSTERING QUALITY ACROSS LAYERS

Figure E.1 shows the clustering quality of features of different layers in different iterations. The cluster assignment quality generally improves with more iterations. In the first iteration, features in

the middle layers show higher quality than the other layers. The target for the first iteration (MFCC clusters) is of worse quality, thus later layers that are more correlated with targets do not yield the best cluster. Target quality improves with more training iterations, thus the best feature layer shifts towards the end. Setting a larger number of clusters increases the clustering quality as can be seen from the comparison between "varied clusters" and "2K clusters". In terms of the 12th layer which we choose, the highest NMI (44.2%) is achieved in the last iteration. In addition, more iterations of training improves the overall quality of clusters produced by a model though the highest NMI/purity among all layers does not necessarily increase in later iterations. Therefore, setting a larger number of iterations brings stable gains which are more robust to the index of layer chosen for clustering. It is important as the purity/NMI, whose measurement rely on a supervised model, are not used for hyperparameter search in practice.

Figure E.1: Quality of feature clusters from different layers across different iterations (BASE, 433 hours unlabeled data). (Iter $i$, Layer $j$): cluster quality of layer-$j$ feature of iter-$i$ model. Upper row: 100, 500, 1K, 2K clusters for 4 iterations. Bottom row: 2K clusters for all iterations. Purity/NMI of the initial MFCC clusters: 30.3%/21.5%

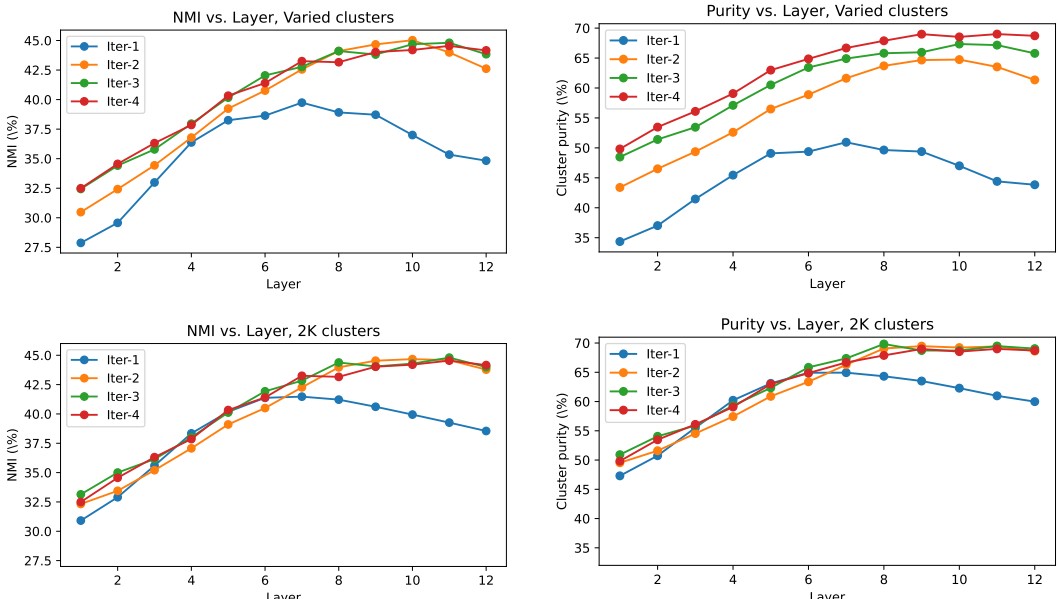

## F    QUALITATIVE EXAMPLES

Figure F.2 shows the example outputs from different models. Our self-supervised model is the LARGE AV-HUBERT pre-trained with 1,759 hours unlabeled data and fine-tuned with 433 hours labeled data. The baseline model is the supervised baseline trained with 433 hours labeled data. Both models use the S2S criterion for supervised training and have the same number of parameters. Qualitatively, our approach provides transcriptions with much higher quality. The baseline approach confuses among words of similar sound while our model output more semantically sound sentences. Figure F.2 also shows typical errors made by our model. We noticed many errors are on short sentences. This is mainly because lip reading relies heavily on the context for recognition due to the existence of homophones. Thus the error rates in lip reading are notably higher in short utterances, which differs from ASR, as can be seen from figure F.1. Substitution among words with homophemes ('fiction' vs. 'vision' in a.4, 'part' vs. 'bunk' in b.5) is another source of error made by the model.

Figure F.1: WER vs. sentence length for lip reading (left) and ASR (right)

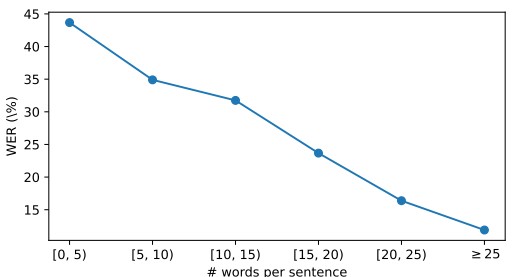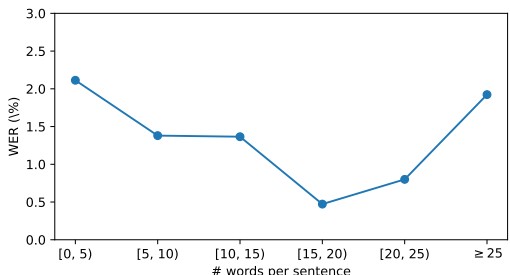

Figure F.2: Transcriptions from different lip-reading models. GT: ground-truth, Proposed: self-supervised model, Supervised: supervised model. Red: wrong words in the output

**(a) Self-supervised vs. Supervised**

(1)
GT: why not ask all of the states to do that instead
Proposed: why not ask all of these things to do that instead
Supervised: why can't i actually all of these things do things and

(2)
GT: indeed we run the risk of making things worse
Proposed: indeed we want the risk of making things worse
Supervised: in india we roughly receive money in the health world

(3)
GT: my desire to disappear was still very powerful
Proposed: my desire to disappear was still very powerful
Supervised: my son is speaking with children about food

(4)
GT: the silent majority does not need to be silent
Proposed: the same majority does not need to be silent
Supervised: this time the total disaster needs to be designed

(5)
GT: mortality is not going down it's going up
Proposed: mortality is not going down it's going up
Supervised: we're seeing this not only carrying slowly how

**(b) Failure cases**

(1)
GT: it's a win all around
Proposed: he's a win on the ground

(2)
GT: sort of leadership by humiliation
Proposed: so the leadership by communication

(3)
GT: is it about equality
Proposed: ask about quality

(4)
GT: science fiction is one of the greatest and most effective forms of political writing
Proposed: stage vision is one of the greatest and most effective forms of political writing

(5)
GT: we can't identify with that part
Proposed: we can't identify with that bunk

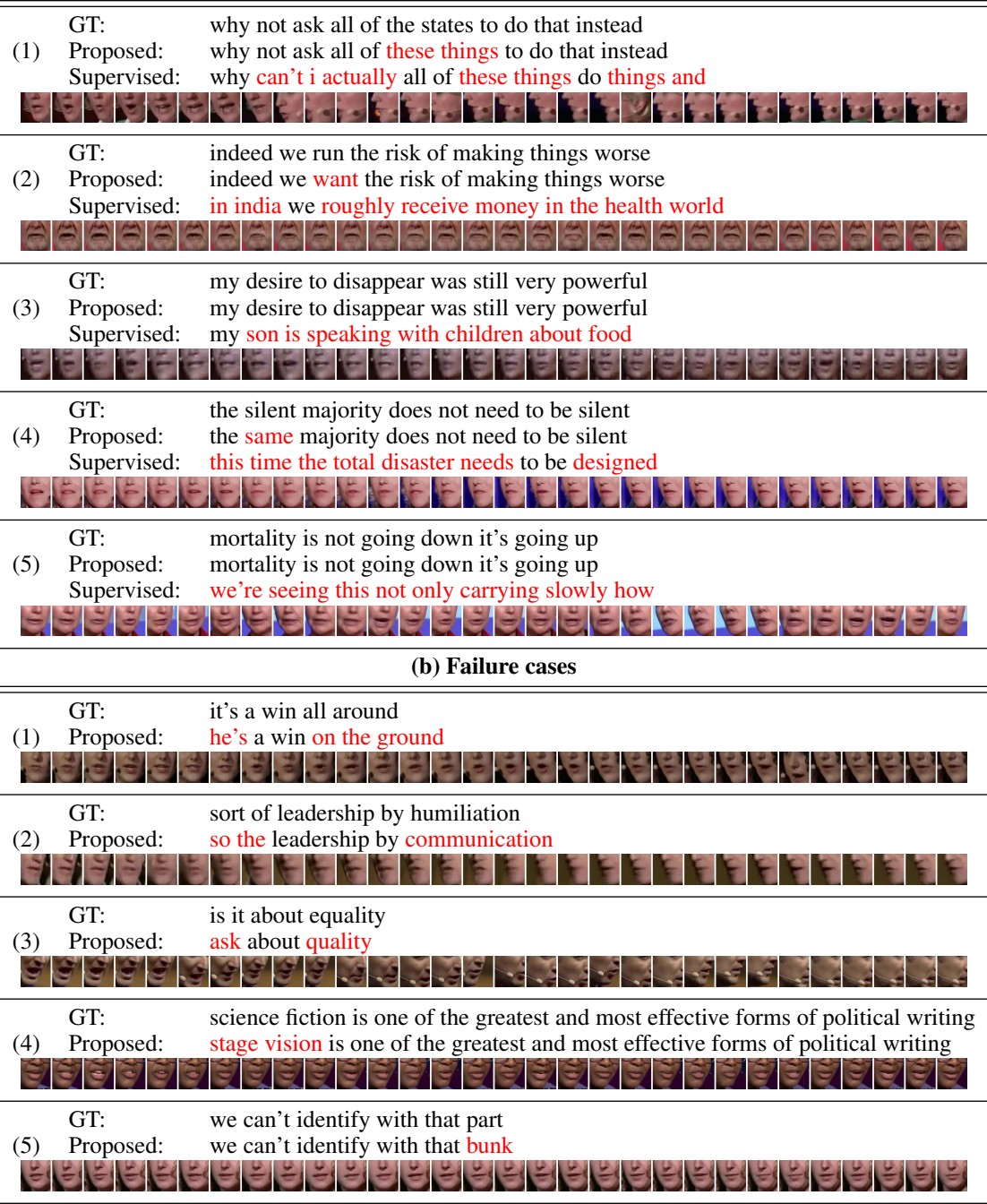

