# OpenReview forum: "Learning Audio-Visual Speech Representation by Masked Multimodal Cluster Prediction"
_ICLR.cc/2022/Conference — ICLR 2022 Poster_

### Official Review · Reviewer_9sxE · 2021-10-30

**Correctness:** 3
**Technical Novelty And Significance:** 4
**Empirical Novelty And Significance:** 3
**Recommendation:** 8
**Confidence:** 5

**Main Review:**

Strength:
1.	The authors propose a powerful method to learn speech representations from multi-modal data which can be utilized for both ASR and VSR, in a self-supervised manner.
2.	AV-HuBERT achieves state-of-the-art performance in terms of WER in a sentence-level audio-visual dataset, LRS3. Moreover, even in a low-resource setting (training with 30 hours labeled data + 1,759 hours unlabeled data), their model shows a comparable VSR result with a previous state-of-the-art method that trained with 31,000 hours labeled data.
3.	Their proposed method is well verified in various views with extensive experiments through both manuscript and supplementary. (Performances 1) with different model sizes, 2) with different loss functions (CTC and S2S), 3) with different cluster targets, 4) using multi-lingual data, 5) of both ASR and VSR, and 6) of ablation studies.)

Weakness:
1.	Even the proposed AV-HuBERT shows impressive performances, but it is an expansion of HuBERT to work in multi-modal data.
2.	There are many typos and errors in the manuscript. Please refer to below detailed comments. Moreover, Figure 1 should be improved, the arrows are arranged confusingly.

Questions:
1. The authors use concatenation as the default fusion operator. Which dimension is utilized for concatenation, temporal or channel? If the temporal dimension is used for multi-modal fusion, better to refer to the two papers [R1, R2].
2. For the loss function in equation (4), their final performance seems obtained by setting the alpha as 0. However, it is hard to find because it is placed in the results of ablation studies in the appendix. Since setting the alpha as zero (the same as omitting the second loss term) is better performed, it would be better to describe the effect of alpha in AV-HuBERT after equation (4) or at the experimental setup, briefly.
3. The ASR performance of AV-HuBERT is not presented. It is just described as the AV-HuBERT performs worse than HuBERT in text. Moreover, the last sentence in Sec 4.5 seems not enough to explain why AV-HuBERT is not better than HuBERT on ASR, even if multi-modal data is utilized during training. Have the authors re-examined hyper-parameters (m_a, m_v, p_m, p_a, alpha) for ASR task?
4. 2nd paragraph in Sec A.4. What does the sentence “Both modalities are used at test time for feature extraction” mean? There is no downstream task utilizes both modalities.

Errors:
1. Sec 3.2 line #4. In “The cluster assignment z_(1:T)^a~”, Does z_(1:T)^a mean z_(1:T)^i?
2. 3rd paragraph on page 4. “When only modality is used”  -> "only one modality".
3. Last paragraph on page 5. “where B consists of all possible...” -> “where B^-1 maps all possible...”. B seems a mapping function thus the word “consist” seems not appropriate.
4. 2nd paragraph on page 6. “we rely on the joint decoder module”. What does the joint decoder mean? Do the authors use a joint CTC/S2S decoder?
5. 2nd paragraph on page 8. ‘iterative pe-training’ -> ‘iterative pre-training’
6. 1st paragraph on page 14. CMUDict (cmu) -> maybe missing reference.

[R1] Chen, Yen-Chun, et al. "Uniter: Universal image-text representation learning." European conference on computer vision. Springer, Cham, 2020.
[R2] Lee, Sangho, et al. "Parameter Efficient Multimodal Transformers for Video Representation Learning." International Conference on Learning Representations. 2020.

**Summary Of The Paper:**

This paper presents a new self-supervised learning framework for audio-visual speech. Their framework is trained with BERT-like training that predicts representations in the masked regions using audio and visual inputs. Extensive experiments are performed using two popular audio-visual datasets. Especially, with a small amount of labeled training data (30 hours), their method achieved comparable results with a previous state-of-the-art method which trained with a huge amount of labeled data (31,000 hours). Moreover, when trained with more labeled data (433 hours), it surpasses the previous state-of-the-art method (trained with 31,000 hours) by 3.1% WER.

**Summary Of The Review:**

Overall, authors approach is promising for reducing the need for a large labeled visual dataset for training VSR models, and the result shown in the paper is significant.

---

> ### Author Response · Authors · 2021-11-16
> **Response to Reviewer 9sxE (Part 1)**
>
> *Q1*: Even the proposed AV-HuBERT shows impressive performances, but it is an expansion of HuBERT to work in multi-modal data.
>
> *A1*: Please refer to our official comments.
>
>
> *Q2*: Figure 1 should be improved, the arrows are arranged confusingly.
>
> *A2*: The source of an arrow refers to the feature used for clustering and generating targets for the model an arrow points to. For example, the second iteration V/MFCC->A model predicts the cluster assignment generated by clustering the first iteration A/MFCC->A features, so there is an arrow point from A/MFCC->A iteration one to V/MFCC->A iteration two. Could you elaborate more about why the arrows are arranged confusingly?
>
>
> *Q3*: The authors use concatenation as the default fusion operator. Which dimension is utilized for concatenation, temporal or channel? If the temporal dimension is used for multi-modal fusion, better to refer to the two papers [R1, R2].
>
> *A3*: Thank you for pointing out the two references and we will add them to the reference. We concatenate along the channel dimension because the audio and visual stream in audio-visual speech data are temporally aligned. In addition, we focus on learning frame-level speech representation. Concatenating along the channel dimension produces frame-level multimodal features, which further improves the quality of labels we use for the next iteration.
>
> [R1, R2] concatenates along the temporal dimension, partially because the two streams are not temporally aligned and this is the only choice.This is similar to many other prior works in multimodal video representation learning mentioned in section 2, which focus on learning sequence-level video representation for high-level downstream tasks such as video/audio classification, image-text retrieval, VQA, etc. The recognition task addressed in this paper generally requires the visual representation capturing fine-grained details in each frame. Thus we think concatenating along the channel dimension per frame would be a more suitable choice for our problem setting.
>
>
> *Q4*: For the loss function in equation (4), their final performance seems obtained by setting the alpha as 0. However, it is hard to find because it is placed in the results of ablation studies in the appendix. Since setting the alpha as zero (the same as omitting the second loss term) is better performed, it would be better to describe the effect of alpha in AV-HuBERT after equation (4) or at the experimental setup, briefly.
>
> *A4*: We will mention setting alpha to 0 in the main text. We keep alpha in the equation mainly to have a comparison with the original HuBERT [1]. As is mentioned in [1], applying prediction on masked frames only is one key ingredient and predicting labels in unmasked frames would greatly degrade its performance. On the contrary,  having a non-zero alpha is much less harmful to learn visual speech representation here as unmasked prediction could also enforce distilling audio-related information to visual stream, as is detailed in section C. Though we did set alpha to 0 in the end, we believe that it is still worthwhile keeping it in general and emphasize its difference with the original HuBERT.
>
> Reference:
>
> [1]. Wei-Ning Hsu, Benjamin Bolte, Yao-Hung Hubert Tsai, Kushal Lakhotia, Ruslan Salakhutdinov, and Abdelrahman Mohamed. Hubert: Self-supervised speech representation learning by masked prediction of hidden units. arXiv preprint arXiv:2106.07447, 2021a.

---

> > ### Author Response · Authors · 2021-11-16
> > **Response to Reviewer 9sxE (Part 2)**
> >
> > *Q5*: The ASR performance of AV-HuBERT is not presented. It is just described as the AV-HuBERT performs worse than HuBERT in text. Moreover, the last sentence in Sec 4.5 seems not enough to explain why AV-HuBERT is not better than HuBERT on ASR, even if multi-modal data is utilized during training. Have the authors re-examined hyper-parameters (m_a, m_v, p_m, p_a, alpha) for ASR task?
> >
> > *A5*: Thank you for pointing this out. We agree that the claims in section 4.5 can be improved. More precisely, we would say that the AV-HuBERT models presented in Table 1 do not outperform a vanilla audio HuBERT model. As a reference, the Base model pre-trained with 1,759 hours unlabeled data in table 1 achieves WER of 4.6/2.4 when fine-tuned with 30h/433h labeled data (it will be mentioned in the updated paper),  which is only slightly better than Base A/MFCC->A (5.0/2.4) and worse than A/MFCC->AV(3.8/2.0). Conceptually the proposed model (A/MFCC-AV)  in table 4 can be regarded as a special case of AV-HuBERT, where p_a is set to 1, p_m is set to 0 and “Fuse” operator is set to “Add”, since it is also trained with AV cluster. We think the relatively worse performance of the same models of table 1 on ASR is probably because the set of hyperparameters, including p_m, p_a, m_a, m_v,  is selected based on the lip reading performance. In the model design, one important aspect we focus on is to distill information from audio to visual representation and prevent the audio input from dominating the pre-trained representations. Thus a stronger visual representation, which leads to lower WER for video only input, probably means more knowledge from audio distilled into it. But there is no guarantee that such a model for audio-only recognition (i.e., ASR) would also be better. However, one clear benefit of AV-HuBERT over audio-only Hubert is its inherent multimodal cluster, which is of higher quality than single-modal clusters (see table D.1). Zeroing out the visual branch for AV-HuBERT, which is named as the A/MFCC->AV, is the most straightforward way of employing such an advantage. A more fine-grained grid search over hyperparameters will probably outperform simply setting p_a to 1 and p_m to 0 (i.e., A/MFCC->AV), which will remain as our future work.
> >
> >
> > *Q6*: 2nd paragraph in Sec A.4. What does the sentence “Both modalities are used at test time for feature extraction” mean? There is no downstream task utilizes both modalities.
> >
> > *A6*: The feature extraction means extracting audio-visual features for clustering, which is used in generating target clusters for the next iteration of pre-training. We will make it clearer.
> >
> >
> > *Q7*: Sec 3.2 line #4. In “The cluster assignment z_(1:T)^a~”, Does z_(1:T)^a mean z_(1:T)^i?
> >
> > *A7*:  Yes. This is a typo. We will fix it.
> >
> > *Q8*: 3rd paragraph on page 4. “When only modality is used” -> "only one modality".
> >
> > *A8*: This is a typo. We will fix it.
> >
> > *Q9*: Last paragraph on page 5. “where B consists of all possible...” -> “where B^-1 maps all possible...”. B seems a mapping function thus the word “consist” seems not appropriate.
> >
> > *A9*: Yes. B is the label collapsing function defined in CTC. We will rewrite this sentence.
> >
> > *Q10*: 2nd paragraph on page 6. “we rely on the joint decoder module”. What does the joint decoder mean? Do the authors use a joint CTC/S2S decoder?
> >
> > *A10*: ”joint decoder module” is the decoder within an S2S model. In using S2S, we do not jointly train or decode with CTC. The wording is a bit misleading and we will fix it.
> >
> > *Q11*: 2nd paragraph on page 8. ‘iterative pe-training’ -> ‘iterative pre-training’
> >
> > *A11*: This is a typo. We will fix it.
> >
> > *Q12*: 1st paragraph on page 14. CMUDict (cmu) -> maybe missing reference.
> >
> > *A12*: It is an issue with the latex bib package. The reference for CMUDict is in the first line of the reference section.

---

> > > ### Comment · Reviewer_9sxE · 2021-11-29
> > > **Thank Authors for the response.**
> > >
> > > I keep the original score with a positive rating. The arrows in the figure 1 are not regularly arranged, which can be confusing to the reader. For example, some arrows disappear after the transformer layer or appear after passing the modality dropout layer. Authors can fix this on the final submission.

---

### Official Review · Reviewer_vjDT · 2021-11-01

**Correctness:** 4
**Technical Novelty And Significance:** 3
**Empirical Novelty And Significance:** 3
**Recommendation:** 6
**Confidence:** 5

**Details Of Ethics Concerns:**

Might be good to address issues regarding the lip reading tasks, e.g. privacy.

**Main Review:**

The ideas are intuitive and very suitable for the problem. The authors take good advantage of a recent advancement of pre-training in ASR (HuBERT), and applies it to lip reading, aka visual speech recognition. The visual-only HuBERT is a simple adaptation, but the cross-modal iterative refinement is a useful addition.

The performance is strong on both low and full resource settings, and the results clearly demonstrate the advantages of self-supervised pre-training.

It would be good to also show the effectiveness of the embeddings for other downstream tasks, such as audio-visual speech recognition and speech separation.

I think Table C1 in the appendix is quite useful for justifying design choices and at least some of it would be beneficial to show in the main paper. It would also be good to show lip reading results after visual-only single-modality pre-training (like audio HuBERT), without cross-modal training.

As an application paper, the paper might be better suited for CVPR or Interspeech, but I still vote for acceptance since the contributions are significant and the results are state-of-the-art.


**Summary Of The Paper:**

The paper proposes a strategy for self-supervised pre-training for lip reading. To this end, they propose Av-HuBERT which learns embeddings by masking video input and predicting iteratively refined hidden units. The authors propose two sub-strategies for this: visual-only and cross-modal. The pre-trained embeddings are fine-tuned on the lip reading tasks, in the style of wav2vec 2.0 and HuBERT for automatic speech recognition. The authors perform experiments on both limited and full resource settings, on both of which they demonstrate strong performance.

**Summary Of The Review:**

The authors propose an effective adaptation of HuBERT for lip reading, by adding cross-modal iterative training. The contributions are logical and the results are strong.

---

> ### Author Response · Authors · 2021-11-15
> **Response to Reviewer vjDT**
>
> *Q1*: It would be good to also show the effectiveness of the embeddings for other downstream tasks, such as audio-visual speech recognition and speech separation.
>
> *A1*: Thank you for your suggestions. The extension to more downstream tasks is our ongoing work. Below are the results we currently have for the two tasks you mention.
>
> For audio-visual speech recognition, we test the model under noise setting where Babble noise is added to audio at different SNR ratios, as is typically done in the audio-visual speech recognition literature [1,2]. Under low-resource setting (30h labeled), our BASE model with audio-visual input achieves 28.7% WER on average in SNR ratios of {-10, -5, 0, 5, 10} dB, as is compared to 70.0% WER without pre-training for audio-visual input.
>
> For speech separation, we focus on employing visual modality to address this task. We simulate this setting by adding a speech audio to the original clip and running our audio-visual speech recognizer to recognize the original speech. Though this problem setting differs from traditional source separation setup, we believe it can indirectly show that the learned representation is beneficial to separate different audio tracks if it can recognize the original speech with high performance. Similar to babble nose setting, we add the interfering speech at SNR ratios varying in {-10, -5, 0, 5, 10} dB. In the low-resource scenario (30h labeled), our BASE model achieves a WER of 13.2%, as is compared to 40.8% (audio-visual input, w/o pre-training), 48.3% (audio input, w/ pretraining) and 66.5% (audio input, w/o pretraining).
> To summarize, the learned representation is also highly effective for audio-visual speech recognition and video-based speech separation.
>
> Reference:
>
> [1]. Triantafyllos Afouras, Joon Son Chung, A. Senior, Oriol Vinyals, and Andrew Zisserman. Deep audio-visual speech recognition. IEEE transactions on pattern analysis and machine intelligence, 2018a.
>
> [2]. Bo Xu, Cheng Lu, Yandong Guo, and Jacob Wang. Discriminative multi-modality speech recognition. In CVPR, 2020.
>
>
> *Q2*: I think Table C1 in the appendix is quite useful for justifying design choices and at least some of it would be beneficial to show in the main paper.
>
> *A2*: We thank the reviewer for the suggestion. The ablation study results are presented in the appendix mainly due to the space constraint, and we will move it to the main paper if the camera-ready version permits additional pages.
>
>
> *Q3*: It would also be good to show lip reading results after visual-only single-modality pre-training (like audio HuBERT), without cross-modal training.
>
> *A3*: We agree with the reviewer that it is crucial to include the performance of visual-only single-modality pre-training like audio HuBERT. In fact, we have included such results in Table 2, which is the “V/HoG $\rightarrow$ V” row: it can be observed that the improvement is very limited (80.3% and 80.1% for the first two iterations) compared to the baseline without pre-training (83.7% in Table B.3) when using only the visual modality. The analysis presented in Section D.1 and Table D.1 explains why: the cluster quality HoG features (1.6% NMI) are terrible compared to the cluster quality of MFCC features (21.5% NMI).
>
>
> *Q4*: Might be good to address issues regarding the lip reading tasks, e.g. privacy.
>
> *A4*: We thank the reviewer for the suggestion. We will strengthen our ethical statement and include such discussions. In particular, our model takes lip region crops as the input, which would make it extremely difficult to infer the speaker identity from the visual stream.
>
>
> *Q5*: As an application paper, the paper might be better suited for CVPR or Interspeech, but I still vote for acceptance since the contributions are significant and the results are state-of-the-art.
>
> *A5*: We thank the reviewer for the comment, but we would also like to emphasize that the proposed AV-HuBERT is more than a straightforward adaptation of the original audio-HuBERT. It tackles a more general problem about self-supervised representation learning from time-synchronous multi-stream sensory data, and demonstrates that even single-modal downstream tasks like lip-reading can significantly benefit from multimodal self-supervised pre-training. Several generalizable techniques are also proposed in this paper to address the input mismatch between multimodal pre-training and single-modal fine-tuning, such as modality dropout and independent random span masking for individual streams.

---

> > ### Comment · Reviewer_vjDT · 2021-11-25
> > **Response to authors**
> >
> > I have read the other reviews and the authors' rebuttals. I am still inclined towards acceptance and maintain my rating.

---

### Official Review · Reviewer_3PrM · 2021-11-03

**Correctness:** 4
**Technical Novelty And Significance:** 3
**Empirical Novelty And Significance:** 3
**Recommendation:** 8
**Confidence:** 4

**Main Review:**

Pros:

a) Simple but very effective extension of masked self-supervised training to multi-modal tasks

b) State of the art lip-reading and ASR results on the LRS3 benchmark

c) Demonstrates value of providing multi-modal input to the model even when only a single modality is present at test time.  Alternatives that consider only that modality as input (image of lips in this case) are significantly worse.

Cons:

Not a negative, but it will be interesting to see further experimentation with layer-wise supervision approaches that inform visual representation learning via audio or AV networks.  This will have the advantage that the model will only see the visual modality at training time, yet will have the advantage of supervision from audio or AV model at all layers, not just at the output.



**Summary Of The Paper:**

This paper presents a multi-modal (audio-visual) pre-training approach that results in models that can be fine tuned to achieve state of the art performance on both lip-reading (visual only) as well as speech recognition (audio only) on the LRS3 dataset which is the largest public lip-reading benchmark. It extends the idea of masked self-supervised learning to multi-modal scenarios where units that are masked and predicted are automatically discovered via clustering and refined as training progresses.  While the proposed approach is a relatively straight forward extension of previously published HuBERT model it is demonstrated to be very effective.  Paper also presents a number of alternative approaches for leaning embeddings of the visual modality and very interestingly demonstrates the value of multi-modal input to the model even when a single modality is present at test time.


**Summary Of The Review:**

Very well written paper, presenting an effective approach for learning embeddings for multi-modal data.

---

> ### Author Response · Authors · 2021-11-15
> **Response to Reviewer 3PrM**
>
> *Q1*: Not a negative, but it will be interesting to see further experimentation with layer-wise supervision approaches that inform visual representation learning via audio or AV networks. This will have the advantage that the model will only see the visual modality at training time, yet will have the advantage of supervision from audio or AV model at all layers, not just at the output.
>
> *A1*: Thank you for your suggestion. It is an interesting idea. For our model, the main reason for having supervision in the last layer is that the target label (i.e. cluster id) is noisy and can be errorful. Predicting only at the final layer allows the model to have intermediate representations that potentially correct this error. As is shown in figure D.1, features in middle layers tend to have higher clustering quality than the last layer in the initial iterations where the target labels are generally of low quality. However, when a generally reliable audio representation (either discrete or continuous) is available, having layerwise supervision would probably help learn a more comprehensive visual representation. Exploring this direction will be our future work.

---

> > ### Comment · Reviewer_3PrM · 2021-11-25
> > **Maintaining rating**
> >
> > I have read author's response and other comments and discussions on this paper and would like to maintain my rating.

---

### Official Review · Reviewer_BjAV · 2021-11-04

**Correctness:** 3
**Technical Novelty And Significance:** 3
**Empirical Novelty And Significance:** 3
**Recommendation:** 6
**Confidence:** 3

**Main Review:**

## Strength:

++ This paper is basically well-written with detailed supplementary.

++ The idea of leveraging Bert for self-supervised training is different from the traditional audio-visual alignment pertaining.

++ Detailed improvements such as the audio-visual stream formulation and Masking by substitution have been proposed.

++ The results on LRSv0.4 outperform the previous SOTA.

## Weakness:

-- This kind of formulation, though novel on the topic of audio-visual speech recognition, has been used on other topics. The whole model is built upon HuBert, which limits the contribution.

-- Why use only the LRS3 dataset for evaluation? The LRS2 dataset should be available. More dataset evaluations could be more comprehensive than one.

-- About the **Masking by substitution**: Why would "the fake segment detection sub-task improves the learned features of the ResNet encoder" when the “filled-in frames are from real videos"? More intuition should be given. It is said that your motivation is to keep "masked region smooth temporally". I understand that it would be smoother than random noise masking, however, segments from the same video cannot guarantee temporal smoothing.

-- I am curious about the "MULTILINGUAL VS. MONOLINGUAL" experiments in Sec. 4.4. The model is kept in a setting with one 1 iteration training and 30h of labeled data. While I understand the limitation in resources, the authors are suggested to show the multilingual results under the full setting. With extra data involved, it might take a longer training time. After all the method is "agnostic to the spoken language".


**Summary Of The Paper:**

This paper proposes audio-visual HuBert based on the original HuBert model. The general idea is similar to HuBert that predicts clusters of extracted features. The authors further propose the modality dropout and masking by substitution module to further fine-tune the design. Extensive experiments and ablation studies are carried out on the LRS3 dataset, which validate the effectiveness of the proposed method.

**Summary Of The Review:**

Overall this paper introduces techniques that the field of audio-visual speech recognition has rarely leveraged into this field, and makes task-specific modifications. The idea is not thoroughly new but the application is good. As I am no expert in this field, it is possible that I have missed something.

---

> ### Author Response · Authors · 2021-11-15
> **Response to Reviewer BjAV (Part 1)**
>
> *Q1*: This kind of formulation, though novel on the topic of audio-visual speech recognition, has been used on other topics. The whole model is built upon HuBert, which limits the contribution.
>
> *A1*: Please refer to our official comments.
>
>
> *Q2*: Why use only the LRS3 dataset for evaluation? The LRS2 dataset should be available. More dataset evaluations could be more comprehensive than one.
>
> *A2*: We would love to evaluate AV-HuBERT on LRS2. However, the license ([link](https://www.bbc.co.uk/rd/projects/lip-reading-datasets)) does not permit usage by all researchers, e.g., doesn’t permit commercial organizations, and hence the LRS dataset is in fact not available to everyone in the research community. Note that such a restriction had also prevented many prior studies [1,2] from evaluating their large-scale experiments on the LRS2 dataset.
> On the other hand, LRS3 is the most widely used benchmark for large-scale lip reading models. Using LRS3 allows us to directly compare our model to existing approaches from other academic and industrial organizations. In addition, LRS3 is the largest publicly available lip reading dataset. Its total duration of videos is twice the size of LRS2, which makes it easy to simulate both low-resource and high-resource settings. Compared to other lip reading datasets, it is also more challenging as all the videos are collected from online resources with more diverse acoustic and linguistic variations. It covers a wide range of speakers (>5K) and has a speaker-independent setup, which is harder yet more realistic.
>
> Reference:
> [1].  Takaki Makino, Hank Liao, Yannis Assael, Brendan Shillingford, Basilio Garcia, Otavio Braga, and Olivier Siohan. Recurrent neural network transducer for audio-visual speech recognition. In Interspeech, 2019.
>
> [2]. Brendan Shillingford, Yannis Assael, Matthew W. Hoffman, Thomas Paine, Cían Hughes, Utsav Prabhu, Hank Liao, Hasim Sak, Kanishka Rao, Lorrayne Bennett, Marie Mulville, Ben Coppin, Ben Laurie, Andrew Senior, Nando de Freitas. Large-Scale Visual Speech Recognition. In Interspeech, 2019.
>
> *Q3*: About the Masking by substitution: Why would "the fake segment detection sub-task improves the learned features of the ResNet encoder" when the “filled-in frames are from real videos"? More intuition should be given. It is said that your motivation is to keep "masked region smooth temporally". I understand that it would be smoother than random noise masking, however, segments from the same video cannot guarantee temporal smoothing.
>
> *A3*: The intuition behind masking by substitution is that it increases the difficulty of the task which in turn  boosts the quality of the learned features . To successfully predict the cluster indices of the original frames for the masked spans, the model has to solve two tasks: identifying which frames are masked, and predicting the label of those based on the context (unmasked frames). The first task --- identifying masked frames --- is made more difficult when the masked spans are replaced with real spans sampled from the same sequence, compared to replacing those with the alternatives studied in Table C.1 (Gaussian noise, learned embedding, spans from other sequences, random frames from the same sequence).
>
>
> To be more specific, in using real images for masking, the model has to rely on fine-grained visual cues such as detecting an abrupt change of mouth shape between two neighboring frames for such distinguishment. The realisticity of masking with segments from the same video comes from three aspects.
>
> 1. An individual masked image frame is visually similar to any unmasked frame as they are both human mouth regions (see Sub vs. Learned Embedding, Sub vs. Gauss. Noise in table C.1). This is the most important factor contributing to the gain of masked by substitution, as can be seen from table C.1.
>
> 2. Sampling from the same video avoids the potential dissimilarity brought by speaker difference (see same seg vs. diff seg in table C.1).
>
> 3. The corrupted sequence is temporally smooth within the masked region as the entire replaced segment is a consecutive chunk. Compared to replacing with independently sampled single images, such smoothness brings additional gains (see sub (same, seg) vs. sub (same, frm) in table C.1).
>
> As pointed out by the reviewer, there still are some inconsistencies around the boundary frames in masked segments and their neighbors. How to further reduce such inconsistency is an interesting question and will remain as future work. We agree that only mentioning temporal smoothness is incomplete and potentially misleading, and will add more detailed explanation in the corresponding section.

---

> > ### Author Response · Authors · 2021-11-15
> > **Response to Reviewer BjAV (Part 2)**
> >
> > *Q4*: I am curious about the "MULTILINGUAL VS. MONOLINGUAL" experiments in Sec. 4.4. The model is kept in a setting with one 1 iteration training and 30h of labeled data. While I understand the limitation in resources, the authors are suggested to show the multilingual results under the full setting. With extra data involved, it might take a longer training time. After all the method is "agnostic to the spoken language".
> >
> > *A4*: Thank you for pointing this out. Actually, we have done experiments of running all the 5 iterations of our model. The test WERs in the final iteration of a BASE model fine-tuned with CTC on 30h/433h are 48.9%/44.0%, while its mono-lingual counterpart achieves 47.3%/43.0%. Indeed, the performance gap between multilingual and mono-lingual models decreases in later iterations. Overall the mono-lingual model still outperforms its multilingual counterpart as shown in section 4.4. This could result from the fact that while the correlation between lip movements and the sound produced is language-agnostic, the masked language modeling part (predicting the label of a masked frame given its context) is still language-dependent. We will mention those results in the paper and clarify what is language agnostic and what is not.

---

> > > ### Comment · Reviewer_BjAV · 2021-11-25
> > > **Response to Authors**
> > >
> > > I have read reviews from other reviewers and the rebuttal. I think most of my concerns have been addressed.
> > >
> > > Overall I think this is a good paper that is recognized by all reviewers. The authors are encouraged to integrate the contents of the rebuttal into the final version of this paper.

---

### Author Response · Authors · 2021-11-15
**Comments and updates**

We would like to thank the reviewers for their  thoughtful feedback and constructive comments. We will address the common questions raised by multiple reviewers in this thread, and respond to the other comments individually in each reviewer’s thread.

*Q*: Novelty

*A*: From a high-level perspective, the core technical novelty of our method is that we show a self-supervised pre-trained multimodal model (audio+visual) would benefit the representation of each modality (audio-only, visual-only). A large volume of prior work [5,6,7,8] on single-modality representation learning addresses the problem by using one modality as input and derives the training target with a secondary modality. In contrast, we directly learn the multimodal representation by using both as input and show that such learned multimodal representation provides a better starting point for single-modal representation than directly distilling from the extra modality. The existing approaches in this direction are focused either on learning representation [2,3] for multimodal tasks (e.g., VQA, video-text retrieval, etc) or on learning a sequence-level single-modality representation [1,4], which are evaluated on high-level semantic tasks such as video or audio classification and does not suit downstream tasks which require fine-grained frame-level representation such as lip reading and speech recognition. Indeed our model is based on audio HuBERT but we extend it to the multi-modal scenario. Directly applying a single-modality HuBERT would not achieve a good result. In the paper, we presented two straightforward ways of using HuBERT in our problem, which are single-modality HuBERT and cross-modality HuBERT. According to our experiments (table 2), single-modality HuBERT (WER: 80.1%) and cross-modality HuBERT (WER: 69.1%) falls behind AV-HuBERT (WER: 58.2%) by more than 10% absolute. In addition to the multimodality nature, the techniques we added to AV-HuBERT are crucial for high performance and have not been proposed in the context of multimodal representation learning literature before. For example, we apply the dropout mechanism (i.e., modality dropout) to prevent one modality dominating the pretraining process and propose masked by substitution to enhance visual representation learning. Overall those two techniques lowers the WER by around 3% (absolute) on the test set, as is shown in our ablation study.

Moreover, below are the main updates we have on experiments. We further show our proposed model is complementary to self-training, a common approach to use unlabeled data. Combining AV-HuBERT and self-training further lowers WER to 28.6% and 26.9% under low(30h) and high(433h) resource setting respectively, which further enlarges the gain over prior SOTA(33.6%) trained with 31K hours of labeled data. The above updates as well as the other revisions based on reviewers’ comments will be added to the paper.

Reference:

[1]. Chen Sun, Austin Myers, Carl Vondrick, Kevin Murphy, Cordelia Schmid. VideoBERT: A Joint Model for Video and Language Representation Learning. ICCV 2019

[2]. Yen-Chun Chen, Linjie Li, Licheng Yu, Ahmed El Kholy, Faisal Ahmed, Zhe Gan, Yu Cheng, Jingjing Liu. UNITER: UNiversal Image-TExt Representation Learning. ECCV 2020

[3]. Weijie Su, Xizhou Zhu, Yue Cao, Bin Li, Lewei Lu, Furu Wei, Jifeng Dai. VL-BERT: Pre-training of Generic Visual-Linguistic Representations. ICLR 2020

[4]. Sangho Lee, Youngjae Yu, Gunhee Kim, Thomas Breuel, Jan Kautz, Yale Song. Parameter Efficient Multimodal Transformers for Video Representation Learning. ICLR 2021

[5]. AJ Piergiovanni, Anelia Angelova, Michael S. Ryoo. Evolving Losses for Unsupervised Video Representation Learning. CVPR 2020

[6]. Humam Alwassel, Dhruv Mahajan, Bruno Korbar, Lorenzo Torresani, Bernard Ghanem, Du Tran. Self-supervised learning by cross-modal audio-video clustering. NeurIPS 2020

[7]. Antoine Miech, Jean-Baptiste Alayrac, Lucas Smaira, Ivan Laptev, Josef Sivic, Andrew Zisserman. End-to-End Learning of Visual Representations from Uncurated Instructional Videos. CVPR 2020

[8]. Pedro Morgado, Nuno Vasconcelos, Ishan Misra. Audio-Visual Instance Discrimination with Cross-Modal Agreement. CVPR 2021

---

### Decision · Program_Chairs · 2022-01-20

**Decision:**

Accept (Poster)

**Comment:**

PAPER: This paper introduces an extension of the HuBERT audio-only model for the audio-visual setting, allowing for self-supervised pre-training of multimodal model which also performs well on the unimodal tasks (lip-reading and ASR). The paper applies the idea of modality dropout to their multimodal pre-training setup and introduce the idea of masking with substitution as a way to improve visual representation learning. A strong aspect of the paper is its experimental section, showing strong improvement for lip-reading tasks, bringing a new state-of-the-art performance. The experiments also show improvement for ASR task.
DISCUSSION: All reviewers seemed to appreciate the experimental results, with new state-of-the-art performance on both unimodal task, when performing multimodal pre-training. The paper does bring some technical novelty, but primarily because of its application to the audio-visual domain. The modality dropout idea was already explored for other audio-visual tasks such as speech-driven face animation (Abdelaziz et al., ICMI 2020), but the idea of “masking by substitution” seems novel and helps learning better visual representations. The authors were able to address many questions and concerns expressed by reviewers. All reviewers took the time to read these responses and acknowledge them.
SUMMARY: This paper brings an interesting extension of the audio-only HuBERT model for the audio-visual setting. The strength of the paper is in its evaluation, with strong performances, establishing many new state-of-the-art results.  All reviewers supported the acceptance of this paper.